# A gain-of-function TPC2 variant R210C increases affinity to PI(3,5)P$_2$ and causes lysosome acidification and hypopigmentation

Qiaochu Wang[1,2,3,8], Zengge Wang[1,2,3,8], Yizhen Wang[1,2,3], Zhan Qi[1,2,3], Dayong Bai[4], Chentong Wang[1,2,3], Yuanying Chen[1,2,3], Wenjian Xu [1,2,3], Xili Zhu[5], Jaepyo Jeon[6], Jian Xiong[6], Chanjuan Hao[1,2,3], Michael Xi Zhu [6], Aihua Wei [7] ✉ & Wei Li [1,2] ✉

Albinism is a group of inherited disorders mainly affecting skin, hair and eyes. Here we identify a de novo point mutation, p.R210C, in the *TPCN2* gene which encodes Two Pore Channel 2 (TPC2) from a patient with albinism. TPC2 is an endolysosome and melanosome localized non-selective cation channel involved in regulating pigment production. Through inside-out recording of plasma membrane targeted TPC2 and direct recording of enlarged endolysosomal vacuoles, we reveal that the R210C mutant displays constitutive channel activation and markedly increased affinity to PI(3,5)P$_2$. Mice harboring the homologous mutation, R194C, also exhibit hypopigmentation in the fur and skin, as well as less pigment and melanosomes in the retina in a dominant inheritance manner. Moreover, mouse embryonic fibroblasts carrying the R194C mutation show enlarged endolysosomes, enhanced lysosomal Ca$^{2+}$ release and hyper-acidification. Our data suggest that R210C is a pathogenic gain-of-function TPC2 variant that underlies an unusual dominant type of albinism.

Albinism is a group of disorders with less or no melanin in skin, hair, and eyes. According to the organs affected, albinism is usually categorized into oculocutaneous albinism (OCA) and ocular albinism (OA). Syndromic albinism (SA) manifests with OCA or OA and is often accompanied with other complications, such as bleeding diathesis, pulmonary fibrosis, and colitis in Hermansky-Pudlak syndrome (HPS)[1–3]. Human color is a common trait documented in ancient times.

The main genes that control eye and skin color are often closely linked and co-evolved under selection pressure[4–7]. In addition, genes mutated in albinism patients often affect visual acuity as most patients present foveal hypoplasia and nystagmus although the underlying mechanism remains elusive[8].

Hundreds of pigmentary genes are involved in melanoblast development originated from the neural crest, melanocyte development,

[1]Beijing Key Laboratory for Genetics of Birth Defects, Beijing Pediatric Research Institute, Beijing, China. [2]MOE Key Laboratory of Major Diseases in Children, Capital Medical University, Beijing, China. [3]Beijing Children's Hospital, Capital Medical University, National Center for Children's Health, Beijing, China. [4]Department of Ophthalmology, Beijing Children's Hospital, Capital Medical University; National Center for Children's Health, Beijing, China. [5]State Key Laboratory of Stem Cell and Reproductive Biology, Institute of Zoology, Chinese Academy of Sciences, Beijing, China. [6]Department of Integrative Biology and Pharmacology, McGovern Medical School, The University of Texas Health Science Center at Houston, Houston, TX, USA. [7]Department of Dermatology, Tongren Hospital, Capital Medical University, Beijing, China. [8]These authors contributed equally: Qiaochu Wang, Zengge Wang. ✉e-mail: weiaihua3000@163.com; liwei@bch.com.cn

melanosome biogenesis and movement, melanin biosynthesis, pigment signaling pathways under resting and stimulated (UV or MSH) conditions[9,10]. At least 20 genes have been identified as the causative genes of albinism in humans, including eight genes in non-syndromic albinism (nSA) and 12 genes in SA[1]. All of these are of either autosomal or X-linked recessive inheritance. Other albinism genes revealed in mice or other animals (e.g., HPS mouse genes *Cno, Vps33a, Rab38/32, Rabggta*) are candidate genes for human OCA or OA[2,3,7,9,10]. Among the nSA genes, *OCA2, SLC45A2,* and *SLC24A5* encode putative ion channels or transporters involved in melanosomal pH regulation and ion homeostasis[11].

Melanosomal pH is a key player in melanin synthesis[12]. The regulatory mechanism of ion homeostasis in the lumen of melanosome is complex and largely unknown. Recently, a gain-of-function (GOF) variant of chloride channel-7 (ClC-7), Y715C, was reported to induce hypopigmented skin and hair with no obvious influence on irises, and the underlying cause appears to involve lysosomal hyper-acidification although it remains to be determined whether this is also true for melanosomes[13]. Two Pore Channel 2 (TPC2) was initially identified as a lysosomal $Ca^{2+}$ release channel that responds to nicotinic acid adenine dinucleotide phosphate (NAADP)[14,15]. It was later found to mediate mainly $Na^+$ conductance when activated by phosphatidylinositol 3,5-bisphosphate ($PI(3,5)P_2$)[16,17]. It was shown recently that TPC2 could indeed conduct both $Ca^{2+}$ and $Na^+$ depending on the agonists administrated[18]. TPC2 has also been localized to melanosomes where it is gated by $PI(3,5)P_2$ and involved in melanosome pH and membrane potential regulation[19,20]. Specifically, whereas the loss-of-function (LOF) of TPC2 alkalized melanosomes and promoted pigmentation, GOF of TPC2 acidified melanosomes and inhibited melanin synthesis[19,20]. Furthermore, a genome-wide study identified two coding variants in TPC2 (M484L and G734E) that are associated with lighter hair color in Northern Europeans[21]. Both variants were later shown to have increased channel function[22]. In addition, L564P was found to be a predominant TPC2 variant that functioned as an essential intramolecular amplifier for blond hair associated with the M484L GOF effect[23]. Thus, evidence is emerging that *TPCN2*, the gene encoding TPC2, may be a causative candidate gene for hypopigmentation when it harbors GOF variants. However, key evidence is still lacking for the *TPCN2* gene to be accepted as a causative gene for albinism.

Here, we describe a Chinese child, carrying a de novo c.628 C > T (p.R210C) variant, who exhibits hypopigmentation in both hair and skin but with normal ocular examinations. Functional characterizations indicate that the TPC2[R210C] mutant is constitutively active and exhibits enhanced sensitivity to $PI(3,5)P_2$. A knock-in mouse mutant homologous to this point mutation (mTPC2[R194C]) shows hypopigmented coat color and reduced melanosome numbers and pigment in the retina in a dominant inheritance pattern. This identifies *TPCN2* as likely a causative gene of an unusual type of hypopigmentation, being a dominant albinism gene with GOF variants.

## Results
### A de novo *TPCN2* variant identified in a child with albinism
The proband was a 7-year-old girl. She exhibited white skin, blonde hair that darkened to brown with age (Fig. 1a). No apparent nystagmus and photophobia were observed in this proband. She had normal vision acuity. Color fundus photography and optical coherence tomography (OCT) showed normal and well-developed macula and fovea (Fig. 1b, c). No additional noticeable symptoms were recorded in this proband. Her parents presented normal pigmentation. Whole exome sequencing (WES) identified that the proband had a de novo mutation (c.628 C > T; p.R210C) of the *TPCN2* gene (Fig. 1d, e). This variant was evaluated as likely pathogenic based on the ACMG guidelines[24]. The homologous sites are R194 in mouse, K224 in zebrafish, and N197 in chicken. Moreover, the corresponding site is F in vertebrate TPC1, while the corresponding site on TPC3 of zebrafish, chicken, and rabbit is also R residue (Supplementary Fig. 1a).

Based on the human TPC2 cryo-EM structures, R210 is located in the S4-S5 linker of the first 6-transmembrane domain of the TPC2 protein (Fig. 1f). Within this linker is an array of positively charged residues, K203, K204, or K207, involved in binding to the hydrophilic head group of $PI(3,5)P_2$. Substitution of any of these residues with a neutral amino acid reduced $PI(3,5)P_2$ binding affinity to a different extent[25]. R210 is three amino acids downstream from K207, with its side chain placed near one of the acyl chains of $PI(3,5)P_2$; however its function has not been reported. From the high-resolution 3D structures of TPC2, including Apo, ligand-bound closed and ligand-bound open states[25], it is evident that R210 undergoes dramatical conformation changes not only upon $PI(3,5)P_2$ binding but also by the transition from the ligand-bound closed to ligand-bound open states, in which the distance between the guanidine group of R210 and the sn-2 carbonyl group (−C=O) of $PI(3,5)P_2$ shortened from ~4.668 Å to ~2.858 Å, representing an increased electrostatic attractive force of ~2.66 folds according to the Coulomb's law (Supplementary Fig. 1b-e). When R210 is mutated to a cysteine, a weaker or repulsive electrostatic force may form between its sulfhydryl (−SH) group and the carbonyl group of $PI(3,5)P_2$ (Supplementary Fig. 1f, g). Therefore, it is of interest to know how the R210C variant affects the affinity of $PI(3,5)P_2$, how it affects the channel activity, and how it impacts the function of the organelles where it resides.

### The TPC2[R210C] variant has an increased affinity to $PI(3,5)P_2$
To address the above questions, we introduced the R210C mutation to a plasma membrane (PM)-targeted TPC2 variant (L11A/L12A) for electrophysiological studies. Using inside-out patches, we recorded the macroscopic current responses of WT- and R210C-TPC2[L11A/L12A] to 1 μM $PI(3,5)P_2$ applied to the cytoplasmic side (Fig. 2a). Surprisingly, R210C presented distinct kinetics from WT TPC2. Firstly, the time to half peak current ($T_{50}$ activation) was 3.3 folds shorter in R210C than WT group (Fig. 2c, e, g). Secondly, the time to half current wash-off ($T_{50}$ wash-off) was 3.7 folds longer in R210C than WT group (Fig. 2c, e, h). Under conditions with equal concentrations of $Na^+$ in the extracellular and intracellular sides of the patch, the current-voltage (I-V) relationships, determined from voltage ramps, were inwardly rectified in both WT and R210C groups (Fig. 2d, f), as previously reported for WT TPC2[25]. However, consistent with the lower expression levels observed based on fluorescence intensity, R210C had lower peak current amplitude than the WT group (Fig. 2i).

To examine the concentration dependence, solutions with increased concentrations of $PI(3,5)P_2$ were applied to the excised patches. R210C exhibited an apparently higher sensitivity to $PI(3,5)P_2$ than WT (Fig. 3a, c), showing a robust response to as low as 1.4 nM of this ligand, and the current reached the maximum at about 12 nM $PI(3,5)P_2$ (Fig. 3c, d). By contrast, the WT channel barely responded to 12 nM $PI(3,5)P_2$, and the current size continued to increase until the concentration reached 1 μM or higher (Fig. 3a, b). The half maximal effective concentration ($EC_{50}$) of R210C was determined to be lowered by 92 folds compared to WT (Fig. 3e), indicating the binding affinity of R210C to the ligand is greatly increased. Thus, the R210C variant of TPC2 likely acts as a GOF mutant to underlie the pathology of hypopigmentation of skin and hair, in agreement with the previous findings that TPC2 variants with mild GOF, M484L and G734E, are associated with lighter skin and hair colors in Northern Europeans[21].

### TPC2[R210C] is constitutively active on the PM and lysosomes
The $EC_{50}$ of R210C for $PI(3,5)P_2$ was 17.6 times lower than that of TRPML1[26]. Such a super high apparent affinity might endow TPC2[R210C] with sensitivity to resting $PI(3,5)P_2$ levels, rendering the mutant channel constitutive active. Interestingly, we noticed that for the majority (>70%) of cells expressing TPC2[L11A/L12A-R210C], it was difficult to form a tight seal in the on-cell mode, but when the patch was excised, the seal improved gradually after tens of seconds with continued

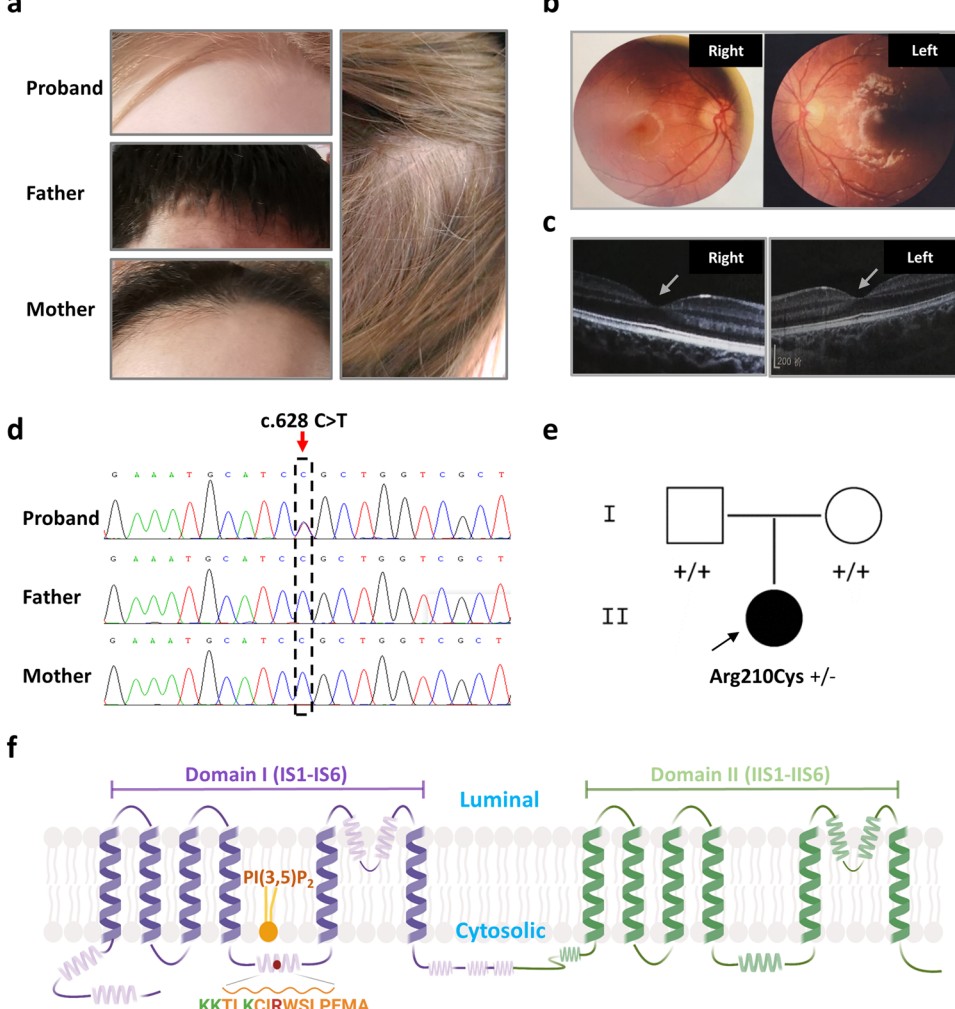

**Fig. 1 | Clinical manifestations and a point mutation of *TPCN2* gene in the proband. a** The proband with hypopigmented skin and hair is shown. Her hair exhibited mixture of white, blond, and brown colors (right panel). Her parents were with normal pigmentation. **b** Normal pigmented retinas and well-developed maculae of the proband. **c** Optical coherence tomography (OCT) examination of the proband presented normal foveae as the arrows show. **d** Histograms of Sanger sequencing data from the proband and her parents revealed a c.628 C > T (p.R210C) mutation (in dashed box) in the *TPCN2* gene of the proband (red arrow). Source data of Sanger sequencing are provided as a Source Data file. **e** A pedigree chart showing the de novo p.R210C mutation of the proband indicated by the arrow. **f** Schematic model of TPC2 secondary structure anchored on a lipid bilayer. TPC2 contains two homologous repeats of six transmembrane helixes and two TPC2 proteins dimerize into one functional channel. The arginine 210 (highlighted in red) is located in the S4-S5 linker helix of the first six transmembrane repeat of TPC2. This motif is characterized by a poly-basic amino acid region containing lysine 203, lysine 204 and lysine 207 (highlighted in green), which interact with the phosphoinositol head of PI(3,5)$P_2$ (highlighted in orange), an agonist of TPC2. The scheme is drawn based on the published structure of TPC2[25] using BioRender with permission. The authors affirm that the patient and her parents provided written informed consent for publication of the images in this figure and the medical information included in this paper.

perfusion (Fig. 4a, red solid line). This phenomenon suggests that TPC2$^{L11A/L12A-R210C}$ localized on the PM was constitutively active, and the activity was dependent on a labile agent(s) easily washed off by perfusion. By contrast, we did not detect discernable basal currents from TPC2$^{L11A/L12A-WT}$ expressing cells either from on-cell patches or immediately after the excision (Fig. 4a, black dotted line). Quantification of current amplitudes at −100 mV from inside-out patches immediately after the patch excision revealed a significant increase in basal current amplitude in R210C compared to WT (Fig. 4b), supporting that TPC2$^{L11A/L12A-R210C}$ is constitutively active, and the activity likely results from a labile endogenous ligand(s).

Given that in inside-out patches, phosphoinositides (PIs) tend to be lost over time with perfusion[27], the agent could be PI(3,5)$P_2$ or other PIs. Although PI(3,5)$P_2$ is not abundantly present on the PM compared to other PIs, such as PI(4,5)$P_2$, PI4P, PI5P, and PI(3,4,5)$P_3$[28], it could be brought to the PM due to the increased lysosomal exocytosis caused by TPC2$^{R210C}$ overexpression. The increased affinity to PI(3,5)$P_2$ (Fig. 3) would then render the PM-localized TPC2$^{R210C}$ active even if only a trace amount of PI(3,5)$P_2$ existed in the PM. Alternatively, the TPC2$^{R210C}$ mutant could exhibit altered selectivity to PIs such that the PI species commonly present in the PM could support its activation. At least, the detection of TPC2$^{L11A/L12A-R210C}$ constitutive activity suggests that it is not competitively inhibited by PIs abundantly present in the PM, as in the case of TRPML1[29].

To examine the selectivity of TPC2$^{R210C}$ for PIs, we applied 1 μM PI(4,5)$P_2$, PI(3,4,5)$P_3$, PI3P, and PI5P to inside-out patches in which currents had been stabilized after the patch excision with continued perfusion and robust responses to exogenously applied PI(3,5)$P_2$ were detected. Interestingly, except for PI(4,5)$P_2$, other tested PI species all evoked sizable TPC2$^{L11A/L12A-R210C}$ currents, although to a much lower extent than PI(3,5)$P_2$ in terms of peak current amplitude. In contrast, only PI(3,5)$P_2$ was able to evoke TPC2$^{L11A/L12A-WT}$ currents (Fig. 4c−g).

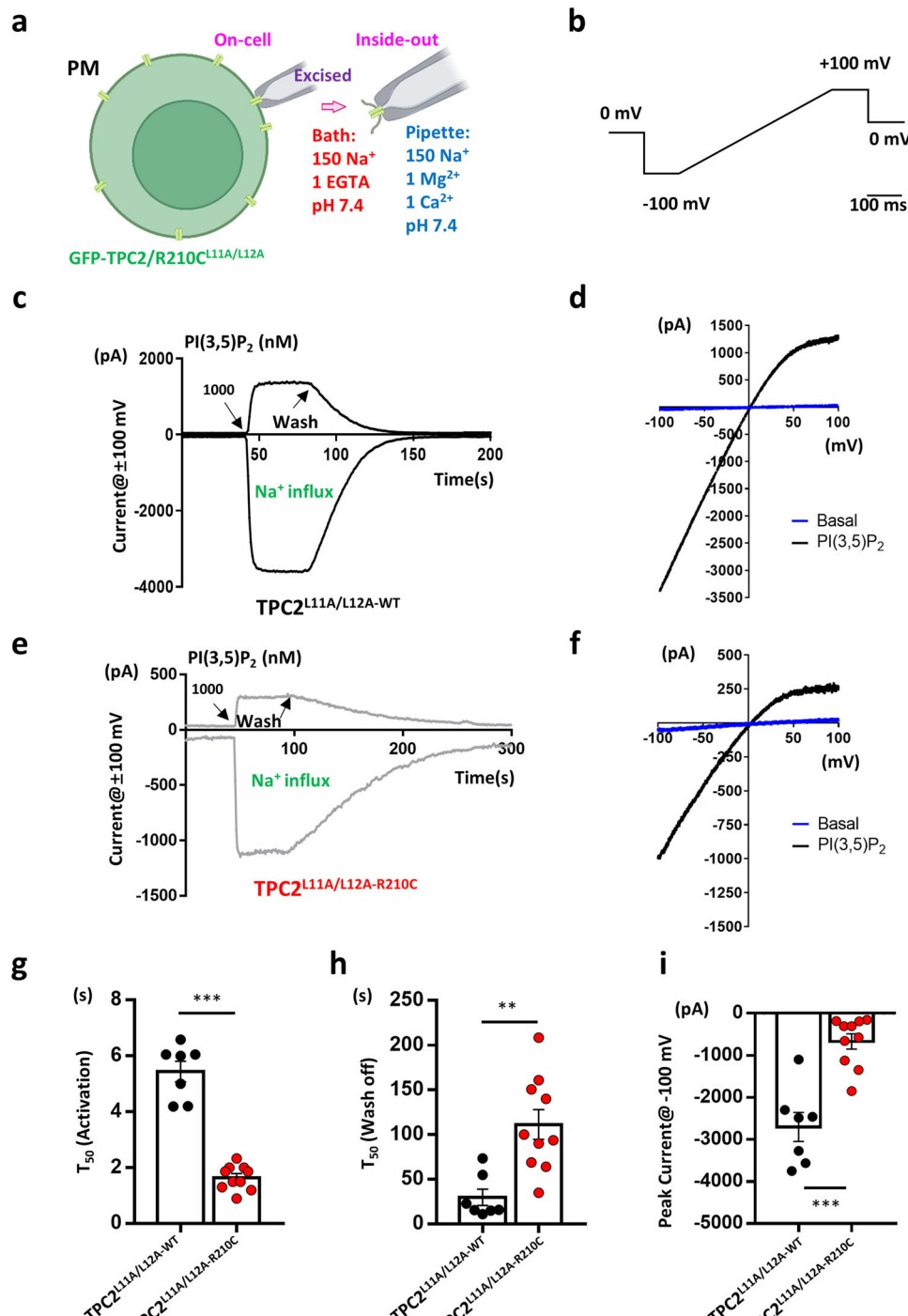

**Fig. 2 | Inside-out recordings of plasma membrane-targeted TPC2 show altered kinetics of TPC2^R210C variant in response to PI(3,5)P₂.** **a** A cartoon depicting the plasma membrane (PM) localization of the TPC2 mutant at the di-leucine motif (L11 and L12 at the N-terminus), TPC2^L11A/L12A, and the patch-clamp configurations using for results shown in Figs. 2–4. The main cation compositions of the pipette solution (in blue) and the bath solution (in red) are listed. Figures 2, 3, and 4c–f show currents recorded in the inside-out period. PI(3,5)P₂-diC8 and other phosphoinositides were applied through perfusion to the cytoplasmic side of the membrane. Figure 4a shows currents recorded in on-cell period, followed by that in the inside-out period. The scheme is drawn using BioRender with permission. **b** Voltage ramp protocol used to elicit ionic currents. **c** Representative traces of TPC2^L11A/L12A-WT currents before and during the application of 1 μM PI(3,5)P₂-diC8 and after the wash-off of the agonist. The upper trace represents outward current at +100 mV reflecting Na⁺ efflux to the extracellular (pipette) side, while the lower trace

represents the current at −100 mV reflecting Na⁺ influx into the cytosol (bath), which is equivalent to lysosomal Na⁺ efflux. Note, only the inward current was used for quantification. **d** Current-voltage (I-V) relationships at basal and the peak of PI(3,5)P₂-evoked response corresponding to (**c**) generated by the voltage ramp protocol shown in (**b**). A bend-over was observed at potentials higher than +40 mV, which is typical for TPC2. **e, f** Representative traces of TPC2^L11A/L12A-R210C currents (**e**) and I-V curves (**f**) showing fast activation but slow wash-off kinetics of the mutant channel as compared to the WT. **g, h** Analysis of time to half maximum current (T₅₀) for PI(3,5)P₂-diC8 induced activation (T₅₀ activation, 5.4 ± 0.4 s for WT, $n = 7$; 1.7 ± 0.1 for R210C, $n = 10$) (**g**) and the time to half current wash-off (T₅₀ wash-off, 29.9 ± 9.2 s for WT, $n = 7$; 111.1 ± 16.7 s for R210C, $n = 10$) (**h**). **i** Analysis of the peak current amplitude at −100 mV evoked by PI(3,5)P₂-diC8. Data in (**g–i**) are presented in mean ± SEM, ∗∗$p < 0.01$, ∗∗∗$p < 0.001$, by Student's unpaired *t*-test, two-tailed. Source data are provided as a Source Data file.

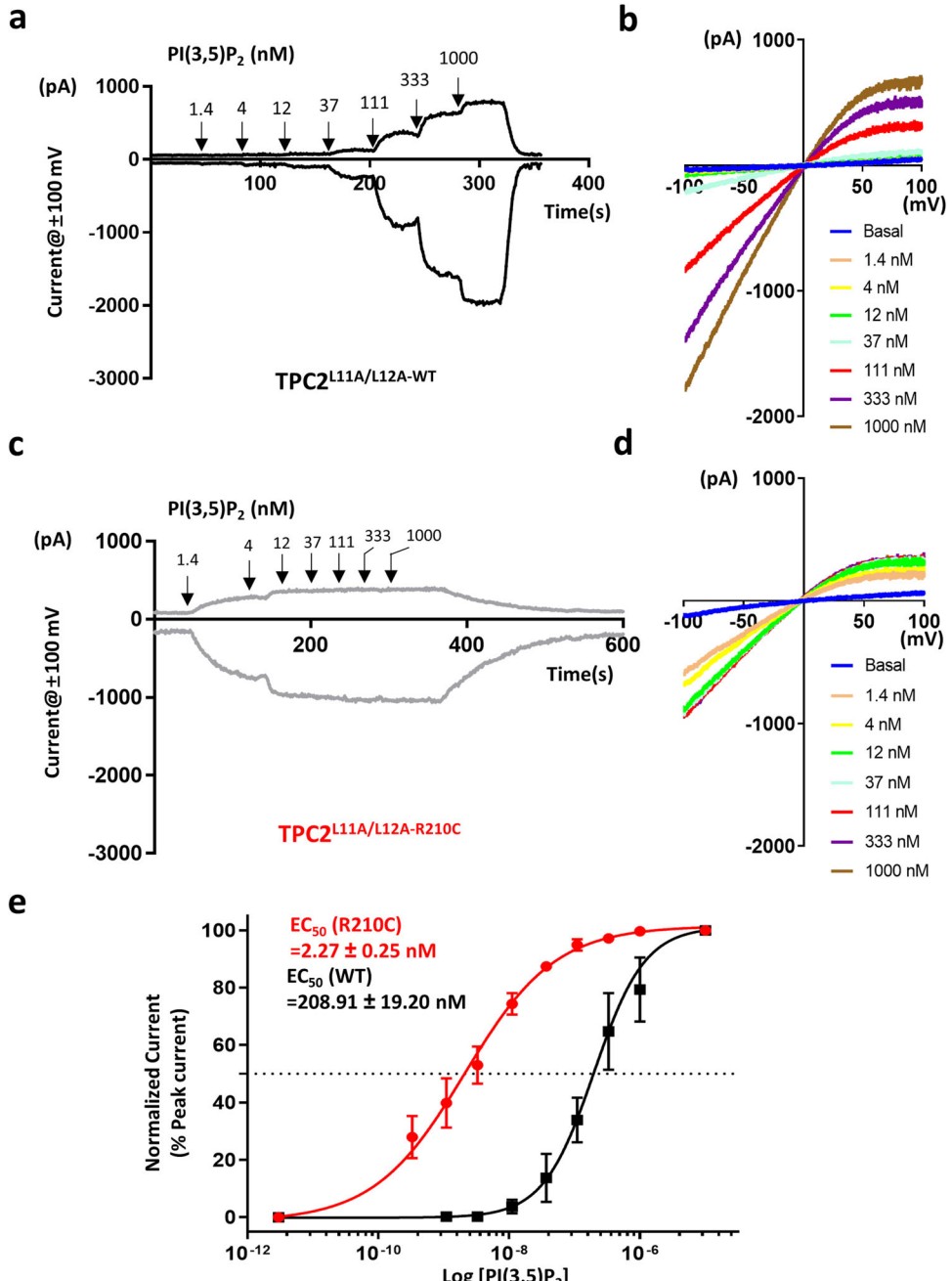

**Fig. 3 | TPC2$^{R210C}$ exhibits increased apparent affinity to PI(3,5)P$_2$.**
**a** Representative traces of TPC2$^{L11A/L12A-WT}$ currents at ±100 mV in response to increasing concentrations of PI(3,5)P$_2$ -diC8 sequentially applied with a threefold increment at the time points indicated by arrows. **b** I-V relationships at the steady states corresponding to **a** for currents evoked by the indicated concentrations of PI(3,5)P$_2$-diC8. **c** Representative traces of TPC2$^{L11A/L12A-R210C}$ currents at ±100 mV in response to the same concentration gradient of PI(3,5)P$_2$ -diC8 as in **a**. Note that 1.4 nM PI(3,5)P$_2$-diC8 elicited large currents, which were absent in TPC2$^{WT}$ group (**a**).

At 12 nM, the PI(3,5)P$_2$-diC8-evoked currents approached saturation. **d** I-V relationships at the steady states corresponding to **c** for current evoked by the indicated concentrations of PI(3,5)P$_2$-diC8. Note the sharp increase with 1.4 nM PI(3,5)P$_2$-diC8. **e** Concentration response curves plotted based on the steady-state currents at −100 mV with varying concentrations of PI(3,5)P$_2$-diC8. Curves are least square fits to the Hill equation. The half maximal effective concentrations (EC$_{50}$) are 208.91 ± 19.20 nM ($n$ = 5) for WT and 2.27 ± 0.25 nM ($n$ = 7) for R210C. Data points are mean ± SEM. Source data are provided as a Source Data file.

Furthermore, there was no difference between the current amplitudes evoked by PI(3,4,5)P$_3$, PI3P, and PI5P, indicating no obvious selectivity on the phosphate groups by R210C. The sensitivity to a broad spectrum of PIs suggests that in addition to the trace amount of PI(3,5)P$_2$, other PIs commonly present in the PM than PI(3,5)P$_2$ could also support the constitutive activation of TPC2$^{R210C}$ on the PM.

To verify if TPC2$^{R210C}$ is also constitutively active in the endolysosomes, we performed whole-endolysosome recording of enlarged vacuoles isolated from vacuolin-1 treated cells that expressed TPC2$^{WT}$

or TPC2$^{R210C}$ (Fig. 5a). We used Na$^+$-Tyrode's as the pipette solution, representing the lysosome lumen, and a K$^+$-based bath solution to represent the cytosol (Fig. 5b). In vacuoles isolated from cells expressing TPC2$^{WT}$, the basal currents were small and the application of PI(3,5)P$_2$ (1 μM) from the bath evoked a robust current increase at −100 mV, but no change at +100 mV (Fig. 5c), consistent with the low permeability of TPC2 to K$^+$. However, in vacuoles isolated from cells expressing TPC2$^{R210C}$, huge currents were observed immediately after forming the whole-endolysosome configuration (Fig. 5d, e). Therefore,

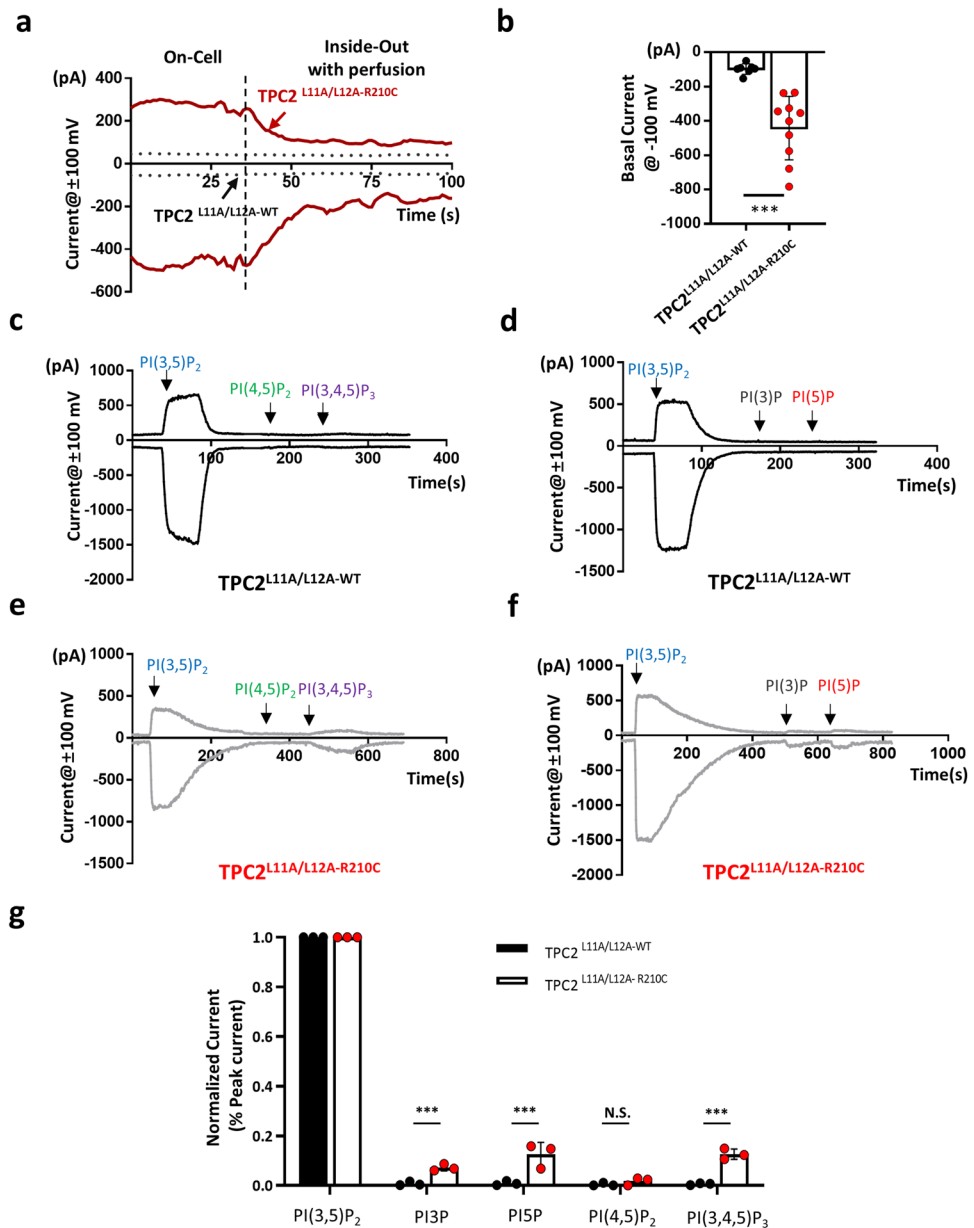

**Fig. 4 | TPC2[L11A/L12A-R210C] is constitutively active on the PM and activatable by additional phosphoinositide species. a** Representative traces of TPC2[L11A/L12A-WT] (gray dotted lines) and TPC2[L11A/L12A-R210C] currents (red solid lines) at ±100 mV recorded in on-cell patches followed by inside-out patches. Recordings were initiated during the on-cell period and continued during and after patch excision (vertical dashed line). Note the gradual decay of the TPC2[L11A/L12A-R210C] currents after patch excision, the time from which the cytoplasmic side of the inside-out patch was continuously perfused with the bath solution. See Fig. 2a for patch clamp configuration and solutions. **b** Analysis of the basal current at −100 mV (WT: −91.9 ± 4.9 pA, $n = 7$; R210C: −478.8 ± 54.5 pA, $n = 10$). Data are presented in mean ± SEM, ***$p < 0.001$ by Student's unpaired $t$-test, two-tailed. Source data are provided as a Source Data file. **c, d** Representative traces of TPC2[L11A/L12A-WT] currents

in inside-out patches during the applications of 1 μM PI(3,5)P$_2$, PI(4,5)P$_2$, PI(3,4,5)P$_3$, PI3P, and PI5P. All PIs showed neglectable current compared with PI(3,5)P$_2$. Black arrows indicate the time points of drug application. **e, f** Similar experiments conducted as in (**c**) and (**d**) but for TPC2[L11A/L12A-R210C]. Except for PI(4,5)P$_2$, all other tested PIs elicited recognizable currents, even though much smaller than that elicited by PI(3,5)P$_2$. Black arrows indicate the time points of drug application. **g** Peak currents at −100 mV normalized to that evoked by PI(3,5)P$_2$. Concentration was 1 μM for all PIs. Each bar represents the currents generated by PIs from three independent experiments. Note, for TPC2[L11A/L12A-R210C], we did not observe differences between PI(3,4,5)P$_3$, PI3P, and PI5P. Data are presented in mean ± SEM, ***$p < 0.001$. N.S., not significant ($p > 0.05$), by one-way ANOVA. Source data are provided as a Source Data file.

similar to the findings from inside-out patches, R210C is also constitutively active in endolysosomal membranes, where PI(3,5)P$_2$ is commonly found[30]. Supporting the idea that resting PI(3,5)P$_2$ levels are involved in the constitutive activation of R210C, the basal R210C current did not decline dramatically until the application of YM201636 (0.8 μM), a specific blocker of PIKfyve[31], a phosphoinositide kinase that produces PI(3,5)P$_2$ (Fig. 5d). Thus, a continued supply of PI(3,5)P$_2$ appears to be maintained by PIKfyve on the endolysosomal membrane to sustain TPC2 activation under unstimulated conditions, and

because of the enhanced sensitivity, the R210C mutant exhibited much higher constitutive activity than the WT channel.

Unexpectedly, all vacuoles isolated from cells expressing TPC2[R210C], but not that from TPC2[WT], showed a significant outward current at +100 mV in the whole-endolysosomal recording. The current was cation-selective as it was eliminated by changing all cations in the bath solution to N-methyl-D-glutamine (NMDG$^+$), a large organic cation impermeable to most cation channels (Fig. 5d). Given that K$^+$ was the main cation in the original bath solution and TPC2[WT] presented

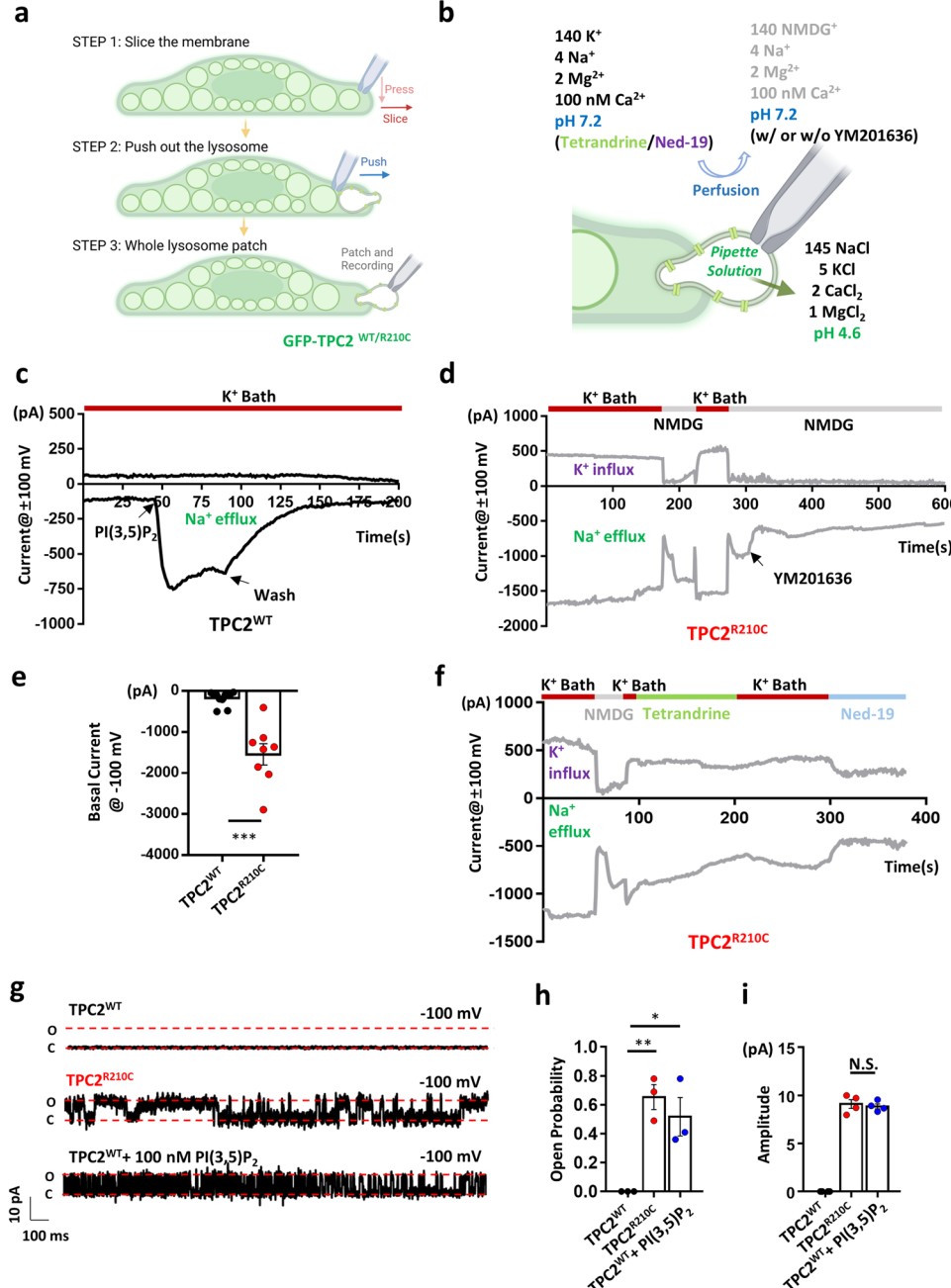

**Fig. 5 | Lysosome-localized TPC2$^{R210C}$ is constitutively active and displays altered ion permeation. a** A cartoon (drawn using BioRender with permission) depicting the three-step protocol used for whole-endolysosome recording from an enlarged endolysosome. **b** A cartoon (drawn using BioRender with permission) depicting the whole-endolysosome configuration and solutions used for inhibiting constitutive currents of R210C in endolysosome patches. **c** A typical TPC2$^{WT}$ response to 1 μM PI(3,5)P$_2$ -diC8. **d** Representative traces of lysosomal TPC2$^{R210C}$ currents in response to NMDG$^+$ and YM201636 (800 nM) as indicated. The outward current at +100 mV carried by K$^+$ was eliminated by substituting all cations in the bath solution with NMDG$^+$. The inward current at −100 mV was sensitive to YM201636, a potent PIKfyve inhibitor that blocks PI(3,5)P$_2$ synthesis. **e** Analysis of the basal current of TPC2$^{WT}$ and TPC2$^{R210C}$ at −100 mV under the resting condition. WT: −49.5 ± 8.7 pA ($n = 11$), R210C: −898.1 ± 346.2 pA ($n = 8$). Data are presented in mean ± SEM, ***$p < 0.001$ by Student's unpaired $t$-test, two-tailed. Source data are provided as a Source Data file. **f** Example traces of R210C currents recorded in whole-endolysosome configuration. NMDG$^+$, tetrandrine (2 μM) and Ned-19

(100 nM) were applied as indicated. Similar results were obtained from three patches. **g** Single-channel activity of TPC2$^{WT}$ and TPC2$^{R210C}$ at −100 mV in inside-out patches excised from HEK293 cells that expressed endolysosomal TPC2$^{WT}$ or TPC2$^{R210C}$. The upper and middle traces were obtained from untreated cells, while the lower trace was obtained from a TPC2$^{WT}$ expressing cell pretreated with 30 μM A1N for 5 min and recorded in the presence of 100 nM PI(3,5)P$_2$ -diC8. For each trace, the closed and open states are labeled with red dashed lines and the total time is 2 s. **h** Bar graph illustrating the mean open probability (Po) of single-channel events of TPC2$^{WT}$ (0, $n = 3$), TPC2$^{R210C}$ (0.65 ± 0.09, $n = 3$), and TPC2$^{WT}$ +100 nM PI(3,5)P$_2$ (0.52 ± 0.13, $n = 3$). Data are presented in mean ± SEM, *$p < 0.05$, **$p < 0.01$, by one-way ANOVA. Source data are provided as a Source Data file. **i** Bar graph illustrating the mean amplitude of single-channel currents of TPC2$^{R210C}$ (8.92 ± 0.36 pA, $n = 3$) and TPC2$^{WT}$ (8.79 ± 0.50 pA, $n = 3$, recorded in the presence of 100 nM PI(3,5)P$_2$). Data are presented in mean ± SEM, N.S., not significant ($p > 0.05$), by one-way ANOVA. Source data are provided as a Source Data file.

negligible outward current representing $K^+$ flux towards the endolysosomal lumen (Fig. 5c), the sizeable outward current recorded from vacuoles expressing TPC2[R210C] suggests that either the R210C mutant had an altered ion selectivity (e.g., enhanced $K^+$ permeability) or the GOF mutant could be functionally coupled to a $K^+$ channel(s) or a nonselective cation channel(s).

We also examined the effect of TPC2 inhibitors, tetrandrine (2 μM) and Ned-19 (100 nM), on the TPC2[R210C] currents in whole-endolysosomes (Fig. 5f). Both drugs were dissolved in the $K^+$-bath solution and applied to the cytoplasmic side (Fig. 5b). While tetrandrine slowly inhibited the inward current of R210C at −100 mV without influencing the outward current at +100 mV, Ned-19 inhibited the R210C currents at both potentials with fast kinetics. This observation suggests that the two blockers act differently on the mutant channel, although they have been reported to be equally effective in inhibiting WT TPC2 and Ebola virus infection[32].

Like TRPML1, the activation of lysosomal TPC2 promotes lysosome exocytosis, which also brings the channel to the PM[18,26]. Consistent with this prediction, the expression of EGFP-TPC2[R210C] (without the PM-targeting mutations) in Hela cells rendered many cells to round up (a sign of cytotoxicity) and displayed colocalization with a PM marker, RFP-CAAX (Supplementary Fig. 2a). Based on the EGFP fluorescence intensity, the PM-to-lysosome ratio of R210C was 7.5 folds of that of the WT TPC2 (Supplementary Fig. 2b). As a control, TPC2[K203A] did not show increased PM localization. To avoid round-up (dying) cells, we captured images from cells expressing low levels of TPC2[R210C] and made 3D reconstruction. Green fluorescence was observed in pseudopods marked by RFP-CAAX in cells that expressed EGFP-TPC2[R210C] but rarely in those that expressed EGFP-TPC2[WT] or EGFP-TPC2[K203A] (Supplementary Fig. 2c).

Supporting the PM localization of TPC2[R210C] and its constitutive activation, we detected single-channel currents with a high open probability of ~0.65 in the absence of any stimulation in inside-out patches excised from HEK293 cells that expressed EGFP-TPC2[R210C] (Fig. 5g, h). For cells that expressed EGFP-TPC2[WT], similar single-channel currents only appeared after a pretreatment of the cells with A1N (30 μM) to bring the channel to the PM, and application of $PI(3,5)P_2$ (100 nM) to the excised patch (Fig. 5g, h). There is no difference in the single-channel current amplitude at −100 mV between TPC2[WT] and TPC2[R210C] (Fig. 5i), indicating that the R210C mutation mainly affects the open probability of TPC2, but not the unitary conductance. Taken together, the R210C mutation acquired enhanced affinity to $PI(3,5)P_2$ and some additional PIs, likely leading to constitutive activation of the channel on the PM and lysosomal membranes.

## *Tpcn2* R194C knock-in mice exhibit hypopigmentation

To verify the pathogenic role of human TPC2 R210C mutant, we generated a homologous mouse *Tpcn2* R194C knock-in strain with CRISPR-Cas9 editing strategy on the C57BL/6J background. The offspring genotypes were confirmed by Sanger sequencing (Supplementary Fig. 3). We observed diluted pigmentation in the fur and tails of the R194C heterozygotes (Het) and homozygotes (Hom), which were viable (Fig. 6a–c). Consistently, we found that the melanin contents in the fur and eye tissues of both Het and Hom were reduced compared with WT, and the fur melanin content in Hom was less than that in Het (Fig. 6d, f, g). Moreover, the number of melanosomes per area in the retinal pigment epithelia (RPE) was less in Het and Hom compared with WT (Fig. 6e, h), but we did not observe substantial change in the choroidal zone of both Het and Hom mice, which indicates that TPC2 mutation may not influence the biogenesis of melanosomes in the choroid (Fig. 6i). Taken together, these results suggest that the R194C knock-in mice recapitulate the hypopigmentation phenotypes as in the human patient in a dominant inheritance manner. The only difference between mouse and human is that the mutant mice presented hypopigmentation in the retinas, mimicking the phenotypes of OCA.

## Lysosomal $Ca^{2+}$ release is increased in *Tpcn2* R194C cells

To understand the pathogenic mechanism of mTPC2[R194C] and its influence on intracellular $Ca^{2+}$ homeostasis, we prepared primary mouse embryonic fibroblasts (MEFs) from the mutant embryos. We performed $Ca^{2+}$ imaging in Fura2-loaded MEF cells for their responses to a recently described synthetic TPC2 agonist, A1N, shown to be PM permeable and able to activate TPC2 efficiently and specifically in intact cells[18]. A nominally $Ca^{2+}$ free extracellular solution was used to exclude $Ca^{2+}$ entry and prevent $Ca^{2+}$ leakage from the cells. Because lysosomal $Ca^{2+}$ release can trigger $Ca^{2+}$ mobilization from the endoplasmic reticulum (ER)[15], we used two concentrations of A1N (10 μM and 30 μM) and checked the ER $Ca^{2+}$ contents with thapsigargin (TG) after the A1N treatment. In WT MEFs, while 10 μM A1N weakly increased cytosolic $Ca^{2+}$ concentration ($[Ca^{2+}]_{cyt}$) without depleting the TG-sensitive ER $Ca^{2+}$ store, 30 μM A1N evoked a robust $[Ca^{2+}]_{cyt}$ rise that rendered the cells no longer responsive to TG (Fig. 7a, f). Therefore, $Ca^{2+}$-induced $Ca^{2+}$ release from ER is triggered by 30 μM AIN, the previously reported concentration[18], but not by 10 μM A1N, in MEF cells. Using these conditions, we compared lysosomal $Ca^{2+}$ release (10 μM A1N) and the combined lysosomal and ER $Ca^{2+}$ release (30 μM A1N) triggered by the membrane permeable TPC2 agonist among WT, Het and Hom R194C MEFs. Hom R194C MEFs responded to 10 μM A1N with faster and greater $[Ca^{2+}]_{cyt}$ increases than Het and WT MEFs, and the Het MEFs also had faster and greater increases than WT MEFs, as shown by both the slope of basal-to-peak ratio and the peak area of the A1N-evoked $Ca^{2+}$ transients (Fig. 7a-c). There were no differences in resting $[Ca^{2+}]_{cyt}$ (Fig. 7d) and TG-induced $Ca^{2+}$ responses (Fig. 7e) among the three types of MEF cells.

When stimulated by 30 μM A1N, the Hom R194C MEFs showed the fastest $[Ca^{2+}]_{cyt}$ rise, as shown by the steeper slope than the Het and WT cells and a shorter delay than the WT cells (Fig. 7f, g). Although the Het R194C MEFs displayed a similar slope in the rising phase of the $[Ca^{2+}]_{cyt}$ increase as WT, they did not show the delay seen in the WT MEFs (Fig. 7f). The area under the trace, which represents the total $Ca^{2+}$ mobilized from the lysosomes and ER, was comparable among these cells (Fig. 7h). Together, these data suggest a sequence order of Hom > Het > WT for the three types of MEFs in the effectiveness of A1N to induce lysosomal $Ca^{2+}$ release, consistent with the R to C mutant being a GOF TPC2 variant for lysosomal $Ca^{2+}$ release channel. Furthermore, the data confirm the dominant effect of the R194C mutation, which is consistent with the coat color changes (Fig. 6d, f).

## R194C MEFs contain more acidified and enlarged lysosomes

Because melanosomes take up fluorescent pH indicator poorly and melanin interferes with the emission of the indicator, we chose to use lysosomes of MEFs as the related organelle to examine intraorganellar pH regulation by mTPC2[R194C]. After staining with LysoTracker Red, a red-fluorescent dye for labeling acidic organelles[14]. The Hom R194C MEFs exhibited more intense fluorescence compared to the WT cells (Fig. 8a, b), suggesting that the R194C mutation resulted in greater acidification of the lysosomes. In addition, we calculated the areas of the individually labeled lysosomes based on their diameters and found that on average, the areas of lysosomes in the Hom R194C MEFs were ~50% larger than that in the WT cells (Fig. 8a, c). Moreover, the increased lysosomal acidification and size enlargement in Hom R194C MEFs were suppressed by Ned-19 (Fig. 8a–c), echoing the effect of Ned-19 in inhibiting the human TPC2[R210C] currents in whole-endolysosomal recordings (Fig. 5f).

Next, we used LysoSensor DND 160 dextran for ratiometric imaging to directly measure endolysosomal pH (Fig. 8d). The pH values for individual lysosomes were determined using the calibration curve (Fig. 8f), which yielded an average pH of 4.11 for WT, 3.85 for Hom, and 5.40 for Hom+Ned-19 (Fig. 8g). Taken together, our data indicate that the R194C mutation leads to enlarged endolysosomal vacuoles and

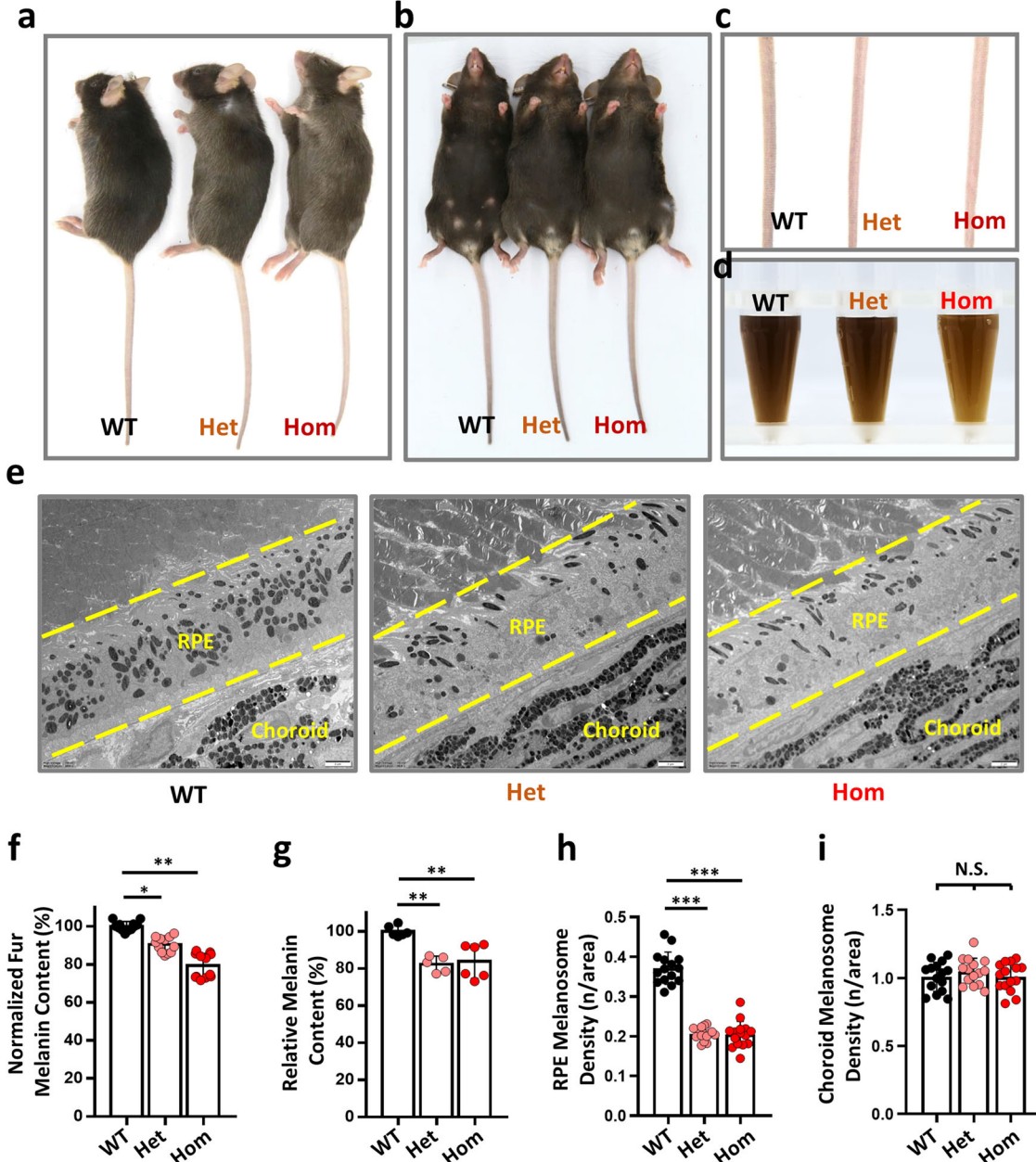

**Fig. 6 | *Tpcn2* R194C knock-in mice exhibit hypopigmentation. a−c** R210C in human TPC2 corresponds to R194C in mouse TPC2 ortholog. The R194C KI mutant was successfully generated by genome editing with a single guide RNA targeting exon 6 of the mouse *Tpcn2* gene. After three generations of heterozygous mating since F1, reduced pigmentation in fur and tails were observed in the heterozygous (Het) and homozygous (Hom) KI mice on the C57/B6J background (WT, wild-type). **d** Side view of fur lysates in tubes. Same amounts (30 mg) of ventral fur from WT, Het and Hom were collected and lysed with a lysis buffer. **e** Transmission electron microscopic pictures of retina dissected from WT, Het and Hom mice. Scale bar, 2 μm. This assay was repeated three times.

**f** Quantitative analysis with a microplate reader revealed that the relative melanin contents in the fur were 90.3 ± 1.2% (Het, *n* = 12) and 79.3 ± 1.8% (Hom, *n* = 12) of that of WT. **g** Relative melanin contents in eye tissues were 82.3 ± 2.0% (Het, *n* = 5) and 83.8 ± 3.8% (Hom, *n* = 6) of that of WT mice. **h** Melanosome densities in the retinal pigment epithelia (RPE) of Het (0.21 ± 0.004, *n* = 15) and Hom (0.20 ± 0.01, *n* = 15) were reduced compared to WT (0.37 ± 0.01, *n* = 15). **i** Melanosome densities in Het (1.05 ± 0.10, *n* = 15) and Hom (1.01 ± 0.10, *n* = 15) choroid were not significantly changed compared to WT (1.01 ± 0.11, *n* = 15). Data in (**f**−**i**) are presented in mean ± SEM, *$p < 0.05$, **$p < 0.01$, ***$p < 0.001$, by one-way ANOVA. Source data are provided as a Source Data file.

hyper-acidification, both of which have been implicated to inhibit melanin synthesis[13,19,20,33].

## Discussion

Although many subtypes of OCA or OA have been molecularly characterized for their causative genes, all of them have been reported as autosomal recessive or X-linked recessive inheritance[1]. We here report an unusual form of albinism with autosomal dominant inheritance both in humans and mice. We identified a de novo variant, R210C, of

the human *TPCN2* gene and proved it to be pathogenic. We found this variant to be constitutively active both on the PM and lysosomes, to have increased apparent affinity to an endogenous TPC agonist, PI(3,5)P$_2$, to exhibit increased Ca$^{2+}$ release in response to TPC2 agonist, to increase [H$^+$] in the lysosomal lumen, and to enlarge the lysosomal volume. As TPC2 also localizes to the melanosome, the GOF mutant may induce similar changes in melanosomes. The decrease of pH within melanosome inhibits melanin production[19,20] and therefore leads to hypopigmentation in hair and skin of both the human patient

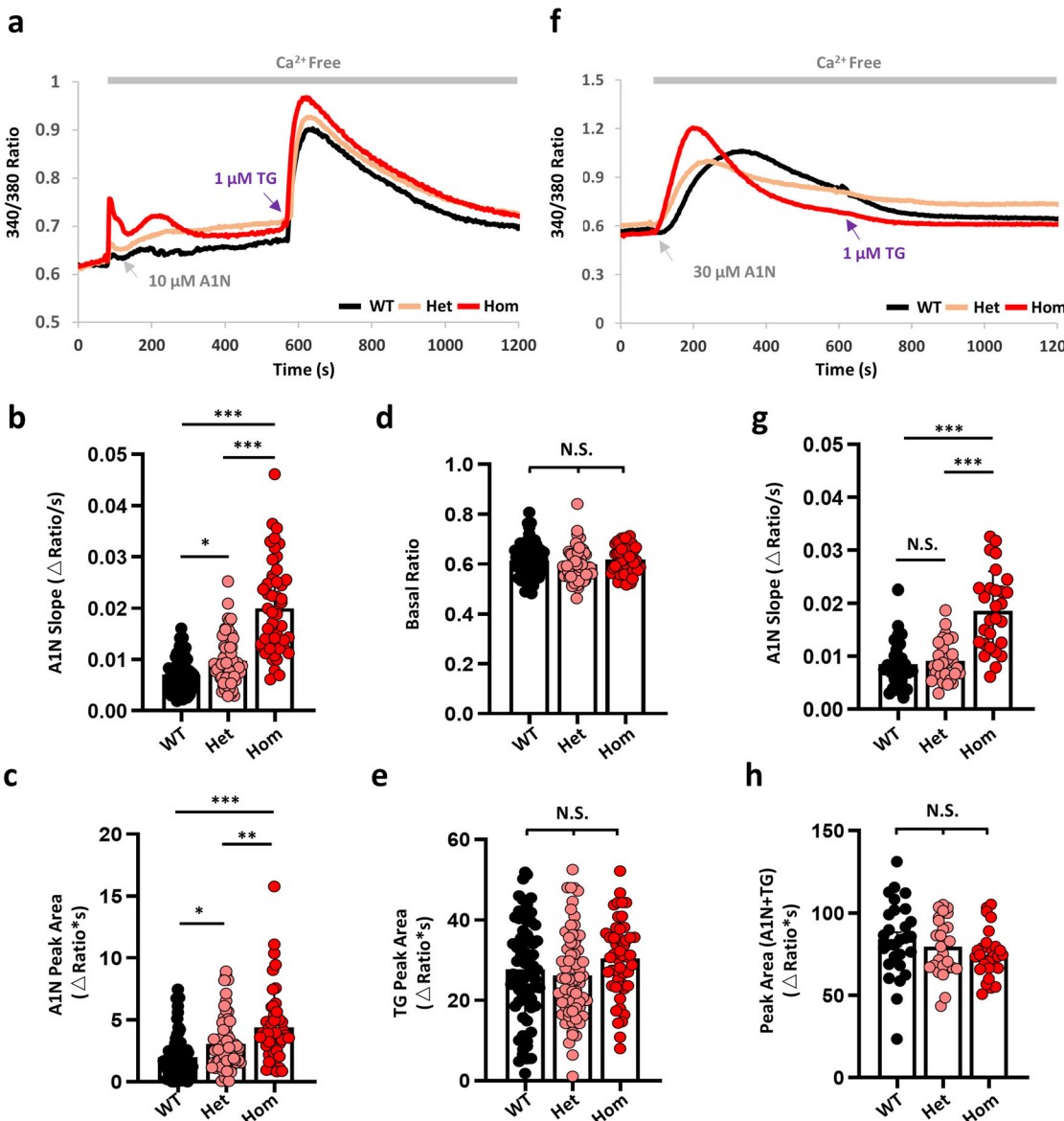

**Fig. 7 | *Tpcn2* R194C knock-in (KI) mouse embryonic fibroblasts (MEFs) show enhanced lysosomal Ca²⁺ release. a** Average ratio (340/380 nm) traces of Fura2 in WT, Het and Hom KI MEF cells in response to sequential treatments of a specific TPC2 agonist, A1N (10 μM, gray arrow) and endoplasmic reticulum (ER) Ca²⁺ ATPase inhibitor, TG (1 μM, purple arrow) in a nominally Ca²⁺-free extracellular solution. **b** Statistics of slope of the initial [Ca²⁺]$_{cyt}$ increase induced by 10 μM A1N, calculated by dividing the peak ratio with the time to reach the peak (in ΔRatio/s). WT: 0.0071 ± 0.0004, $n = 74$; Het: 0.0098 ± 0.0006, $n = 67$; Hom: 0.0200 ± 0.0013, $n = 50$. **c** Statistics of peak area (PA) for the response to 10 μM A1N between 100 s and 580 s (in ΔRatio*s). WT: 2.00 ± 0.19, $n = 74$; Het: 3.06 ± 0.25, $n = 67$; Hom: 4.39 ± 0.41, $n = 50$. **d** Statistics of basal Fura2 ratio averaged from the first ten ratios. WT: 0.613 ± 0.008, $n = 74$; Het: 0.599 ± 0.008, $n = 67$; Hom: 0.618 ± 0.008, $n = 50$.

**e** Statistics of PA for the response to TG between 580 s to 1200 s (in ΔRatio*s). WT: 27.8 ± 1.4, $n = 74$; Het: 26.2 ± 1.4, $n = 67$; Hom: 30.4 ± 1.4, $n = 50$. **f** Average ratio (340/ 380 nm) traces of Fura2 in WT, Het and Hom KI MEF cells in response to the sequential treatments of 30 μM A1N (gray arrow) and 1 μM TG (purple arrow) in the nominally Ca²⁺-free solution. **g** Statistics of slope of the initial [Ca²⁺]$_{cyt}$ increase induced by 30 μM A1N, calculated as in (**b**, in ΔRatio/s). WT: 0.0085 ± 0.0009, $n = 28$; Het: 0.0092 ± 0.0007, $n = 28$; Hom: 0.0186 ± 0.0014, $n = 28$. **h** Statistics of PA for the response to both A1N and TG between 90 s and 1200 s (in ΔRatio*s). WT: 84.1 ± 4.3, $n = 28$; Het: 79.5 ± 3.2, $n = 28$; Hom: 74.8 ± 2.7, $n = 28$. Data in (**b–e**, **g**, **h**) are presented in mean ± SEM, *$p < 0.05$, **$p < 0.01$, ***$p < 0.001$, N.S., not significant ($p > 0.05$), by one-way ANOVA. Source data are provided as a Source Data file.

and the mutant mice bearing the equivalent R to C mutation, as well as that in the eyes of the mutant mice due to the GOF in the mutant TPC2 channel. Thus, the *TPCN2* gene is the causative gene for a unique subtype of cutaneous albinism.

Neutral melanosomal pH is critical for melanin biosynthesis[11,12]. However, the maintenance of neutral pH in the lumen of mature melanosomes involves complex mechanisms and is incompletely understood. Luminal H⁺ homeostasis is maintained and regulated by other ions, such as Cl⁻, Ca²⁺ and Na⁺, and more interestingly by PI(3,5) P₂[13,20,34]. V-ATPase is the proton pump that drives H⁺ entry into

melanosomes; other ion channels or transporters regulate the efficiency of V-ATPase via adjusting the membrane potential and providing counter ions with or without the control of PI(3,5)P₂[20].

Genetic evidence suggests the involvement of a number of ion channels and transporters in melanosome pH regulation. SLC45A2 is a solute carrier responsible for H⁺ efflux from melanosomes. Thus, deficiency of SLC45A2 in OCA-4 patients leads to lower melanosomal pH and inhibition of melanin production[35]. ClC-7 is a Cl⁻/ H⁺ antiporter localized in lysosome membranes and probably melanosome membranes as well. ClC-7 imports two chloride ions into the organellar lumen

in exchange for the efflux of a proton[36]. The net Cl⁻ accumulation in the lumen provides counter ions that help sustain H⁺ entry, promoting luminal acidification, which in turn inhibits melanin production[13]. Unlike our proband, patients who carried the GOF Y715C mutation of ClC-7 exhibited delayed myelination, delayed development, and organomegaly, in addition to hypopigmentation[13]. However, most patients with recessive LOF *CLCN7* mutations exhibit osteopetrosis[37]. Thus, although GOF mutants of both TPC2 and ClC-7 showed hypopigmentation, the GOF Y715C (ClC-7) mutant is apparently more severe than the GOF R210C (TPC2) mutation. It is interesting that the GOF ClC-7 Y715C knock-in mice on the C57BL/6J background showed hypopigmentation[13], while mice with LOF ClC-7 on an agouti background had a gray coat color[38]. These results may be explained by the inhibition of eumelanin production in lower pH in the GOF mice and the inhibition of pheomelanin synthesis in higher pH in the LOF mice. However, no clear evidence has shown that ClC-7 is localized to the melanosomes to regulate its luminal pH. OCA2 is a Cl⁻-selective ion channel for Cl⁻ efflux from melanosomes, which hyperpolarizes the melanosome membrane potential and thereby reduces the V-ATPase activity, opposing the function of ClC-7 in luminal Cl⁻ balance[33]. Thus, LOF OCA2 variants mimic the GOF ClC-7 variants in manifesting hypopigmentation. The above findings support that melanosomal pH is regulated by Cl⁻ homeostasis.

Similar to the anions, cation homeostasis is also important for melanosomal pH regulation. TPC2 affects melanosome pH possibly by depolarizing the membrane potential through releasing Ca²⁺ and Na⁺, thereby enhancing V-ATPase activity and lowering melanosomal pH[20]. Thus, whereas TPC2 knockdown or knockout in melanocytes increased melanosome pH and melanin content, TPC2 overexpression decreased melanosome pH and melanin content[19,20]. Lysosomal Ca²⁺ content has long been proposed to be maintained by a Ca²⁺-ATPase[39], but to date, neither its molecular identity nor its role in melanosomes is known. Alternatively, NCKX5/SLC24A5, the mutation of which causes OCA-6[40], may mediate Ca²⁺ transfer from mitochondria to melanosomes with an unknown mechanism[41]. Deficiency in SLC24A5 leads to disrupted PMEL fibril formation[41], probably by reducing melanosomal pH secondary to lowering melanosomal Ca²⁺.

Interestingly, overexpression of TPC2 in melanocytes counteracted the increased pigmentation caused by OCA2 overexpression[20]. This suggests that TPC2 and OCA2 may coordinate to regulate melanosome pH, raising an important question that homeostasis of melanosomal luminal Cl⁻, Ca²⁺, and/or Na⁺ may be crucial for maintaining melanosome pH. This opens an avenue to address how the ion transporters or channels coordinate in regulating melanosomal ion homeostasis and whether OCA2, SLC45A2, SLC24A5, TPC2, and ClC-7 interact in regulating melanosomal pH and pigment production, which is important to uncover the underlying mechanisms of albinism.

Recent Cyto-EM structures show that the poly-basic motif of TPC2 is critical for PI(3,5)P₂ binding, a notion supported by that mutations in K203, K204, and K207 markedly increased the EC₅₀ of PI(3,5)P₂ on TPC2 activation[25]. In contrast, our study revealed that R210, which is within the PI(3,5)P₂-binding pocket, when mutated to a cysteine profoundly decreased the EC₅₀ of PI(3,5)P₂ by ~92 folds. Unlike K203, K204, and K207, which interact with the polar head of PI(3,5)P₂, R210 is situated near the fatty acyl tails of the phospholipid ligand[25]. PI(3,5)P₂ induces channel activation in at least two steps: (1) it binds to TPC2 to cause a 135° counterclockwise turn of R210 so that the guanidine group faces the sn-1 fatty acyl chain of PI(3,5)P₂ (Supplementary Fig. 1b), and (2) the PI(3,5)P₂-bound closed state undergoes a further conformation change to transit to the PI(3,5)P₂-bound open state[25]. During the second step, not only does R210 further rotate counterclockwise by 90° but also the two fatty acyl chains of PI(3,5)P₂ flip such that the sn-2 chain is now juxtaposed to the S4-S5 linker with its carbonyl group situated close enough to the guanidine group of R210 to form an electrostatic interaction (Supplementary Fig. 1c, d). More importantly, in the ligand-bound open state, the sn-1 fatty acyl tail of PI(3,5)P₂

extents all the way into the hydrophobic region between the S3 and S4 TM segments, whereas in the ligand-bound closed state, it is the sn-2 fatty acyl chain that interfaces with the S3 and S4 segments with its tail bent back towards the S4-S5 linker (Supplementary Fig. 1c). Therefore, the flip of the fatty acyl chains and the consequent extension of the sn-1 fatty acyl tail to the central hydrophobic regions of the S3 and S4 TM segments may be critical for gating transition. In the ligand-bound closed and ligand-bound open states, R210 guanidine is -4.668 Å and -2.858 Å from the carbonyl groups of the sn-1 and sn-2 chains, respectively (Supplementary Fig. 1d). The electronegativity of the oxygen atom of the polar carbonyl group attracts the positively charged guanidine. Here we use an "Attraction and Repulsion" model to explain the fine-tuning mechanism of the 210 site on TPC2. The attractive forces between the guanidine and carbonyl groups may either retard the fatty acyl chain flip in the closed state or constrain the hydrophobic interactions of the fatty acyl tails with the channel in the open state. In either case, the substitution of the guanidine by an -SH group of C210 would change the electrostatic attraction to repulsion between position 210 on TPC2 and the fatty acyl chains of PI(3,5)P₂, making either the flip easier (a faster $k_{on}$) or the hydrophobic interactions more stable (a slower $k_{off}$). Future studies are needed to distinguish these possibilities. Consistent with the above idea, the R210A substitution of human TPC2 also reduced the EC₅₀ of PI(3,5)P₂ (to $19.42 \pm 0.74$ nM. $n = 3$, not shown in figures), but the effect was not as strong as the R210C mutation. Among the three vertebrate TPCs, TPC1 has a phenylalanine (F) at the equivalent position of TPC2 R210 (Supplementary Fig. 1a) and the reported EC₅₀ values for rabbit TPC1 and TPC2 were 26 nM and 208 nM, respectively[42]. Thus, the residue at the position equivalent to 210 of human TPC2 strongly influences TPC gating by PIs, with the positive charge having a negative impact and the negative charge or polarity having a positive effect on the apparent affinities. The mechanism of this tuning effect warrants further investigations.

Although we cannot rule out that the constitutive activity of TPC2^R210C occurred independently of the increased affinity to PIs at this point, the likelihood that the dramatically enhanced affinity allows the basal levels of PI(3,5)P₂ or other PIs to keep the channel active in unstimulated cells is quite high. The robust increase in the apparent affinity of TPC2^R210C to PI(3,5)P₂ may "cash in" on the basal PI(3,5)P₂ to exert effects on pH regulation and trafficking of acidic organelles, which likely underlies the inhibition of melanin production in melanosomes although direct evidence from melanosomes is still needed.

The R210C mutation also showed increased sensitivity to several other PIs except for PI(4,5)P₂, acquisition of K⁺ permeability and mislocalization to the PM. The constitutively active TPC2^R210C on the PM and intracellular organelles disrupts Na⁺ and K⁺ balance and disturbs membrane potential, which could have profound effects to the cell. In the organism, the enhanced sensitivity to PI(3,5)P₂ and the capability to be activated by other PIs may render of TPC2^R210C to play pathological roles in releasing more Ca²⁺ from melanosomes and lysosomes, which disrupts the ion homeostasis of these organelles. Nevertheless, whether there exist pathophysiological changes in other tissues of the TPC2^R210C patient (or TPC2^R194C in mice) requires further careful examinations.

## Methods
### Subjects
The proband, a 2.5-year-old Chinese girl, was initially admitted by the Department of Dermatology, Beijing Tongren Hospital, Capital Medical University (CMU). She revisited at 7 years old. She was diagnosed as albinism according to "Clinical Practice Guidelines for Albinism"[43]. Blood samples were collected from the proband and her parents. This study was approved by the ethics committees of Beijing Children's Hospital and Beijing Tongren Hospital, CMU. The authors affirm that the parents of the patient provided written informed consent to

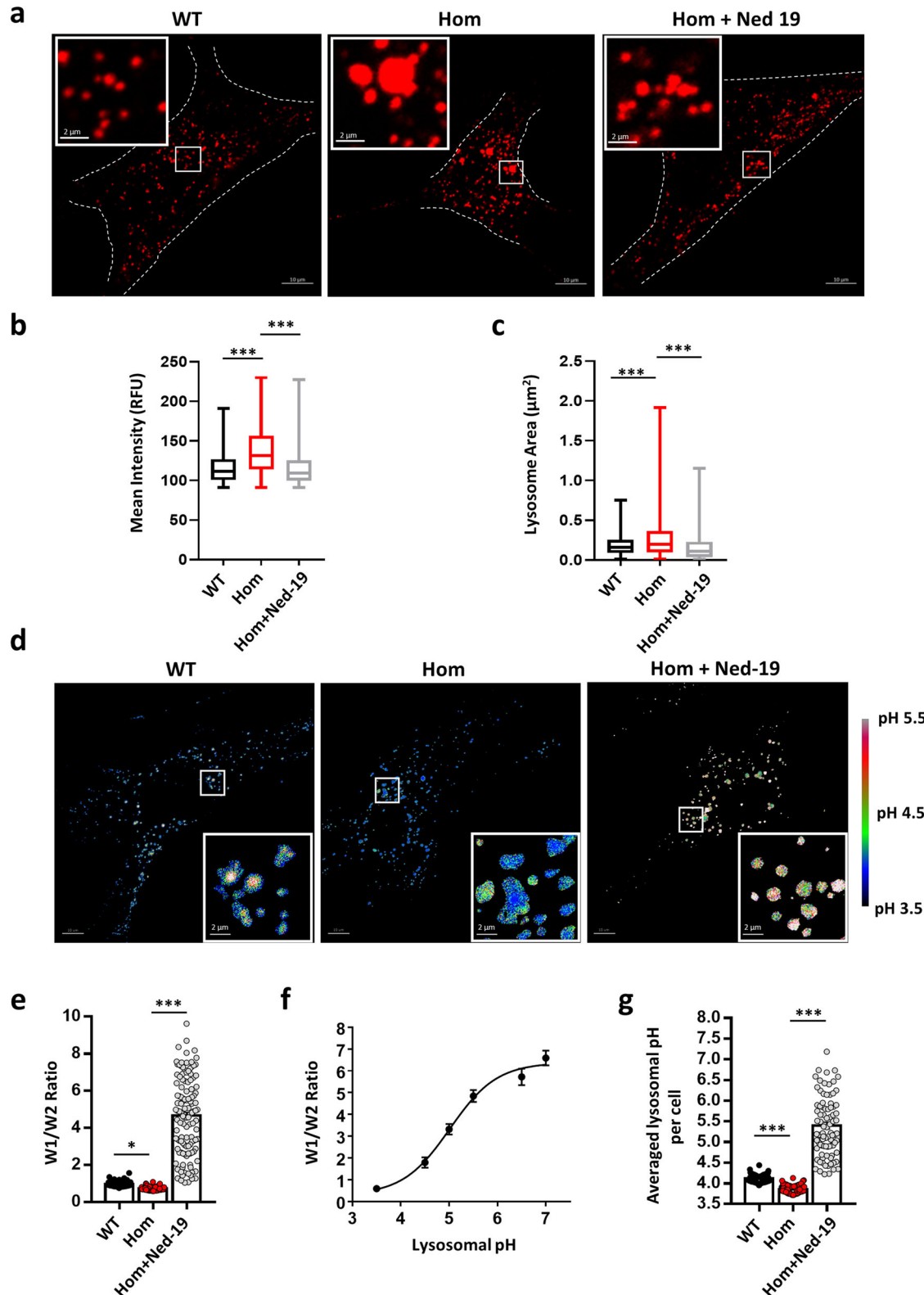

participate in the study, and for the publication of the images in Fig. 1 and the medical information included in this paper. The study was performed in compliance with the principles set out in the declaration of Helsinki.

### Genetic analysis

Genomic DNA was extracted from blood samples by the routine proteinase K/SDS method. Exome sequencing was performed on an Illumina Solexa HiSeq 2000 instrument according to Illumina's protocol (Illumina, San Diego, CA)[40,44]. The pathogenicity of the unidentified mutation was classified according to the ACMG (American College of Medical Genetics) guidelines[24]. PCR amplification was performed with a forward primer: 5′-GCAGAGCATGTGGCCTGGAA −3′, and a reversed primer: 5′- AACCAAAGAGACCCCAAAAGCC −3′. Sanger sequencing of PCR products was performed on ABI PRISM 3700 automated sequencer (Applied Biosystems, Foster City, CA).

**Fig. 8 | *Tpcn2* R194C knock-in MEFs display hyper-acidification and enlarged vacuolar lysosomes. a** Representative images of MEFs stained with LysoTracker Red. WT and Hom MEFs were used. For some Hom MEFs, Ned-19 (100 nM) was applied together with LysoTracker Red. Scale bars in main images: 10 μm; in the insets: 2 μm. **b** Statistics of average intensity (in RFU, random fluorescence unit) of LysoTracker fluorescence. Lysosomes were recognized by using the particle function of Image J software with the same threshold. The minimum, maximum, median, first quartile, third quartile for WT ($n = 1281$): 91.00, 191.2, 118.10, 100.50, 126.90; for Hom ($n = 1216$): 91.00, 229.90, 136.70, 114.30, 156.30; and for Hom + Ned-19 (100 nM, $n = 1038$): 91.00, 227.40, 117.20, 99.66, 125.70. Data were collected from 10 cells from each group seeded in three independent dishes. **c** Statistics of average area (μm²) of lysosomes recognized by LysoTracker Red. The minimum, maximum, median, first quartile, third quartile for WT ($n = 1281$): 0.010, 0.750, 0.200, 0.089, 0.256; for Hom ($n = 1216$): 0.010, 1.914, 0.293, 0.100, 0.365; and for

Hom + Ned-19 (100 nM, $n = 1038$): 0.010, 1.154, 0.162, 0.039, 0.230.

**d** Representative ratiometric (W1/W2) H⁺ images for WT and Hom MEFs and Hom MEFs treated with Ned-19 (100 nM). A rainbow lookup table on the right shows the calibration. Scale bars in main images: 10 μm; in the insets: 2 μm. This assay was repeated three times. **e–g** Statistics of average LysoSensor ratio (**e**) and lysosomal pH (**g**) determined using the calibration curve (**f**). Lysosomes were recognized by threshold and selection function and ratiometric images were generated by image calculator function of Image J software. The W1/W2 ratios **e** are: WT, $1.280 \pm 0.010$ (**n** = 117); Hom, $0.950 \pm 0.009$ ($n = 110$), and Hom + Ned-19 (100 nM), $5.05 \pm 0.19$ ($n = 124$). The lysosomal pH values (**g**) are: WT, $4.11 \pm 0.01$ ($n = 117$); Hom, $3.85 \pm 0.01$ ($n = 110$), and Hom + Ned-19 (100 nM), $5.40 \pm 0.08$ ($n = 124$). Data were collected from 10 cells from each group seeded in three independent dishes. Data in (**e**, **g**) are presented in mean ± SEM, $*p < 0.05$, $***p < 0.001$, by one-way ANOVA. Source data in (**b**, **c**, **e–g**) are provided as a Source Data file.

## Cell culture

HEK293 (CTCC-003-0015) and Hela (CTCC-001-0006) cells were purchased from Meisen CTCC (Chinese Tissue Culture Collections). Cells were certified without any mycoplasma. Cells were maintained at 37 °C, 5% $CO_2$ in Dulbecco's modified Eagle medium (DMEM, Gibco) supplemented with 10% fetal bovine serum (FBS, Gibco) and penicillin-streptomycin (100 U/mL-100 μg/mL, Gibco). Primary mouse embryonic fibroblasts (MEFs) were isolated from E14.5 mouse embryos and cultured following the previous method[45]. Briefly, pregnant mouse was sacrificed. Internal organs and heads of the embryos were removed. Tails were used for DNA extraction and genotyping. The remaining embryonic tissue was cut into small pieces and then dissociated by 0.25% trypsin at 37 °C for 20 min to obtain a single-cell suspension. Isolated MEFs were seeded on Petri dishes with DMEM containing 10% FBS supplemented with penicillin-streptomycin (200 U/mL−200 μg/mL). All cells were trypsinized and passaged every 2 to 3 days when they reached 80% to 90% confluence.

## Chemicals and drugs

Chemicals used for electrophysiological and imaging experiments were purchased from Sigma-Aldrich, MERCK. The lipid ligands including phosphatidylinositol-3,5-bisphosphate- diC8 (PI(3,5)P₂-diC8), PI(4,5)P₂-diC8, PI(3)P-diC8, PI(5)P-diC8, and PI(3,4,5)P₃-diC8 were all purchased from Echelon Biosciences. LysoTracker Red DND-99 (Thermo Fisher Scientific, L7528), LysoSensor Yellow/Blue Dextran (Invitrogen, L22460), TPC2-A1-N (MCE, HY-131614), trans Ned-19 (MCE, HY-103316A), YM201636 (MCE, HY-13228), Fura2 acetoxymethyl ester (Fura2 AM, F14185, Invitrogen), and Tetrandrine (MCE, HY-13764) were purchased as indicated in parentheses.

## cDNA constructs

The expression plasmid of human TPC2 was modified from the N-terminal mCherry-tagged hTPC2 in pIRES-zeocin vector as described[46], but with the coding sequence of mCherry replaced by that of EGFP. The N-terminal EGFP-tagged TPC2^L11A/L12A in pcDNA3 plasmid was originally provided by Dr. Jiusheng Yan (MD Anderson Cancer Center). For both plasmids, the sequences coding for hTPC2 were corrected by site-directed mutagenesis using a PCR-based approach such that the derived protein sequence matches exactly to that of the patient, except for the R201C substitution. To achieve high fidelity, the KOD-Plus-Neo Kit (Toyobo Co. Ltd., Japan) was used for PCR. R210C and the K203A mutations were also introduced by site-directed mutagenesis. The RFP-CAAX plasmid was made by replacing the coding sequence of GCaMP6s with that of tagRFP in the pGP-CMV-GCaMP6s-CAAX vector (Addgene, #52228, gift of Dr. Tobias Meyer)[47]. All constructs were verified by Sanger sequencing.

## Electrophysiology

Recordings were acquired using a MultiClamp 700B amplifier (Axon Instruments) with the low-pass analogue filter set to 2 kHz.

Currents were sampled at 10 kHz using a Digidata 1440 A digitizer (Axon Instruments) and further analyzed with pClamp10 and Clampfit 10.4 software (Molecular Devices). Patch pipettes were pulled from borosilicate glass (BF150-75-10, Sutter) with a micropipette puller (P-97, Sutter Instrument) and heat polished with a microforge (MF2-LS1, Narishige). All current-voltage (I-V) curves of TPC2 and its mutants were obtained using voltage ramps from −100 to +100 mV over a 500-ms duration.

To record TPC2 current on plasma membrane, HEK293 cells expressing TPC2^L11A/L12A-WT and TPC2^L11A/L12A-R210C were used. After the pipette attached to the cell membrane, a giga seal was formed by releasing the positive pressure or a gentle suction from the pipette holder. Inside-out patches were made by pulling the pipette away from the cell surface. The extracellular (pipette) side is equivalent to the luminal side of TPC2 in endolysosomes. The standard bath (cytosolic) solution contained: 145 mM sodium gluconate (Na-Glu), 5 mM NaCl, 1 mM EGTA, 10 mM HEPES, pH 7.4, adjusted by HCl. The pipette (extracellular) solution contained: 145 mM Na-Glu, 5 mM NaCl, 1 mM $MgCl_2$, 1 mM $CaCl_2$, 10 mM HEPES, pH 7.4, adjusted by HCl. These solutions followed that of previously reported[25]. Various concentrations of PI(3,5)P₂ as specified in each experiment were diluted in the bath solution and applied to the cytoplasmic side through perfusion.

To record TPC2 current on enlarged endolysosomal vacuoles, HEK293 cells expressing EGFP-TPC2^WT and EGFP-TPC2^R210C were treated with 1 μM vacuolin-1 overnight before experiments. The procedure followed the published protocol[16,48]. Similar to whole-cell recording, seal was broken by negative pressure and zap with an electrical pulse. The bath solution (cytosolic) contained 140 mM K-gluconate, 4 mM NaCl, 1 mM EGTA, 2 mM $MgCl_2$, 0.39 mM $CaCl_2$, 20 mM HEPES (pH adjusted with KOH to 7.2; free [Ca²⁺] 100 nM). The pipette (luminal) solution was a modified Tyrode's solution with low pH: 145 mM NaCl, 5 mM KCl, 2 mM $CaCl_2$, 1 mM $MgCl_2$, 10 mM glucose, 10 mM HEPES, and 10 mM MES [2-(4-morpholinyl) ethanesulfonic acid hydrate] with the pH adjusted to 4.6 with HCl.

To obtain single-channel currents of TPC2^WT and TPC2^R210C, HEK293 cells expressing EGFP-TPC2^WT and EGFP-TPC2^R210C were used to for inside-out recording. The pipettes had resistances between 10 MΩ and 13 Ω after polishing. The bath solution (cytosolic) contained 140 mM K-gluconate, 4 mM NaCl, 1 mM EGTA, 0.39 mM $CaCl_2$, 20 mM HEPES (pH adjusted with KOH to 7.2; free [Ca²⁺]ᵢ = 100 nM). The pipette (extracellular) solution was a modified Tyrode's solution with neutral pH: 145 mM NaCl, 5 mM KCl, 2 mM $CaCl_2$, 10 mM glucose, and 20 mM HEPES, with the pH adjusted to 7.2 with HCl. Recordings were made using a MultiClamp 700B amplifier (Axon Instruments) with the low-pass analogue filter set to 1 kHz. Currents were sampled at 10 kHz in a Gapfree mode using a Digidata 1440 A digitizer (Axon Instruments) and further analyzed with Clampfit 11.2. The data were base line corrected and single-channel events were identified automatically by the single-channel search function. Unitary current amplitude was obtained from the fit of the amplitude histogram with Gaussian

equation. Open probability (NP$_O$) was obtained through events statistics function of the software. To obtained GFP-TPC2$^{WT}$ single-channel current, cells plated on the cover-glass were pretreated with 30 μM A1N in the bath solution for 5 min before the inside-out recording and 100 nM PI(3,5)P$_2$ included in the bath solution during the recording. Without the combined A1N and PI(3,5)P$_2$ treatment, patches excised from cells expressing EGFP-TPC2$^{WT}$ did not show obvious single-channel activity and therefore served as a good negative control.

### Genomic editing of mice by CRISPR/Cas9 technology

The CRISPR/Cas9 technology was used to introduce the *Tpcn2* R194C variant, homologous to the human *TPCN2* R210C variant, into the C57BL/6J mice. In brief, Cas9 mRNA and gRNA (5′-AGAAGA CCCTGAAGTGTATACGG-3′) were obtained by in vitro transcription, and synthesized oligo donor DNA was obtained from Shanghai Model Organisms Center, Inc. The gRNA targeted mouse *Tpcn2* exon 6, where the oligo donor DNA would introduce the substitution into the mouse genome. Fertilized eggs were collected from C57BL/6J females. Cas9 mRNA, gRNA, and oligo donor DNA were microinjected into the fertilized eggs. All mice studies were approved by the Institutional Animal Care and Use Committee of Institute of Genetics and Developmental Biology, Chinese Academy of Sciences. Genotyping was performed in all generations of mice by PCR followed by Sanger sequencing. DNA was extracted from mouse tails with a DNA extraction kit according to the manufacturer's instructions (Macherey-Nagel, Germany). PCR amplification was performed with the forward primer: 5′-ATGAGCACA GCCTAGGAGGA −3′, and the reversed primer: 5′-AGCTGAGAGAAGC AAGGTTGA −3′.

### Calcium imaging

The ratiometric Ca$^{2+}$ indicator Fura2 AM was used according to our previous protocol[49]. Briefly, cells were incubated with 4 μM Fura2 AM and 0.02% F-127 (w/v) at room temperature (20–22 °C) for 40 min in a Ca$^{2+}$ containing extracellular solution (ECS: 140 mM NaCl, 5 KCl, 2 mM CaCl$_2$, 1 mM MgCl$_2$, 10 mM glucose, and 20 mM HEPES, pH = 7.4). Cells were rinsed twice and kept in the ECS for another 10 min for dye de-esterification. Images were captured on a Nikon inverted microscope (Eclipse Ti). Cells were alternatively excited at 340 and 380 nm, using a single-band multi-exciter filter set (Fura2-C-000, BrightLine®, Semrock), and emission was collected through a 409 nm edge single-edge dichroic beam splitter (BrightLine®, Semrock) and a 510/584 nm single-band emitter (BrightLine®, Semrock). Ratioimages were taken with an EMCCD camera (iXon DU897, Andor) in a time-lapse mode with 3 s intervals. Images were obtained after background was subtraction. Data analysis was conducted in Excel (Microsoft 365) and GraphPad Prism 7.0.

### Lysosomal pH measurement

The protocol for lysosomal pH measurement followed the protocol[20] with minor modifications. MEF cells plated on glass bottom dishes (Cellvis, China) were incubated in culture medium supplemented with 1 mg/ml LysoSensor Yellow/Blue Dextran overnight followed by chasing for 4 h before imaging. Cells were imaged in the Ca$^{2+}$ containing ECS on an LSM 880 confocal scanning microscope (Zeiss, Germany). Pinhole was wide open to acquire more light and broader z-depth. LysoSensor was excited at 405 nm and its emission detected at 417–483 nm (W1) and 490–530 nm (W2). The W1/W2 ratio was used to calculate pH based on a calibration curve generated with solutions containing 125 mM KCl, 25 mM NaCl, 7 μg/ml monensin, 7.5 μg/ml nigericin and varying concentrations of MES by adjusting the pH to 3.5, 4.5, 5, 5.5, 6.5, or 7 from three independent experiments. When needed, 100 nM trans Ned-19 was added in the medium during chasing 1 h before imaging. Data analysis was conducted in Image J and GraphPad Prism 9.0.

### Fluorescent confocal microscopy

For colocalization analysis, Hela cells seeded on 35 mm glass bottom dishes were transfected with the desired plasmids for 8 h with Lipofactamine 3000 (Thermo Fisher Scientific), followed by another 16–24 h of incubation for protein expression. For lysosome staining, MEFs were seeded on glass bottom dishes one day before the experiment. Fully attached cells were rinsed with ECS and incubated with 100 nM LysoTracker Red dissolved in ECS at 37 °C for 1 h. When needed, trans Ned-19 (100 nM) was added together with the 100 nM LysoTracker Red. Images were captured with an LSM 880 confocal scanning microscope (Zeiss) or a dragonfly dual spinning disk microscope (Andor).

### Melanin content measurement

The fur of 8-week-old mice was removed from the dorsal side using an electric clipper and collected into 1.5 mL tubes. For each sample, 30 mg of fur were lysed in 0.3 mL 1 M NaOH solution at 85 °C for 4 h. Eye tissues were dissected and placed in cold phosphate-buffered saline on ice, and then transferred into 1.5 mL tubes with 0.3 ml 1 M NaOH solution. The tissue homogenate was lysed at 56 °C for 1 h. The absorbance at 475 nm of the diluted lysates was measured in a 96-well plate using a plate reader (Multiscan, Thermo).

### Transmission electron microscopy

Tissues were fixed with 2.5% (v/v) glutaraldehyde in a phosphate buffer (PB) (0.1 M, pH 7.4), washed twice in PB and twice in deionized water. Then the tissues were immersed in 1% (wt/v) OsO$_4$ and 1.5% (wt/v) potassium ferricyanide aqueous solution at 4 °C for 2 h. After washing, the tissues were dehydrated through graded alcohol (30%, 50%, 70%, 80%, 90%, 100%, 100%, 8 min each) into pure acetone for two times, 8 min each. Samples were infiltrated in graded mixtures (3:1, 1:1, 1:3) of acetone and SPI-PON812 resin [21 ml SPI-PON812, 13 ml dodecenylsuccinic anhydride (DDSA) and nadic methyl anhydride (NMA)], followed by pure resin. Finally, the tissues were embedded in pure resin with 1.5% N,N-dimethylbenzylamine (BDMA) and polymerized for 12 h at 45 °C and then 48 h at 60 °C. Ultrathin sections (70 nm thick) were cut with a microtome (Leica EM UC6), double-stained by uranyl acetate and lead citrate, and examined by a transmission electron microscope (FEI Tecnai Spirit 120 kV).

### Statistical analysis

Unpaired Student's *t*-test and one-way ANOVA were used to determine statistical significance, using GraphPad Prizm 7.0 or 9.0. Hill equation was applied to fit concentration responses with the use of Origin 8.0 software. Experiments were performed with at least three independent biological replicates. All error bars depict SEM. All the cartoons in the figures were completed with BioRender under publication permissions.

### Reporting summary

Further information on research design is available in the Nature Portfolio Reporting Summary linked to this article.

## Data availability

All data supporting the findings of this study are available from the corresponding authors upon reasonable request. The source data underlying Figs. 1–8 and Supplementary Fig. 2 are provided as a Source Data file. The structures of TPC2 were retrieved from the Protein Data Bank (PDB) using accession codes 6NQ1 (Apo TPC2) and 6NQ2 (ligand-bound TPC2). Source data are provided with this paper.

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

## Acknowledgements

We thank Zhongshuang Lv, Xueke Tan, and Xixia Li for help with electron microscopy sample preparation at the Center for Biological Imaging, Institute of Biophysics, Chinese Academy of Sciences (CAS). We thank Shi-Wen Li at the Institute of Zoology (CAS) for her help in patch clamp experiments. This work was partially supported by grants from the Ministry of Science and Technology of China [2019YFA0802104 (W.L.)], the National Natural Science Foundation of China [82173447(A.W.); 92054102 (Q.W.); 31830054 (W.L.)], the Beijing Natural Science Foundation [5212005 (Q.W.)], and the State Key Laboratory of Molecular Developmental Biology, CAS [O76003N491 (Q.W.)].

## Author contributions

W.L., Q.W., A.W., M.X.Z., and C.H. conceived and designed the study. A.W. and D.B. made the diagnosis and follow-up. Q.W., Z.W., Y.W., C.W., Y.C., X.Z., J.J., and J.X. generated, analyzed, and interpreted the data. Z.Q. analyzed the sequencing data and provided genetic counseling. W.X. provided bioinformatic support. W.L., Q.W., and Z.W. wrote the manuscript. M.X.Z. critically revised the manuscript. All of the authors approved the final version of the manuscript.

## Competing interests

The authors declare no competing interests.
