## [Peer Review File · Nature Communications]

A Gain-of-Function TPC2 Variant R210C Increases Affinity to PI (3,5)P2 and Causes Lysosome Acidification and HypopigmentationReviewers' Comments:

Reviewer #1:

Remarks to the Author:

In this study, Wang et al report on a de novo missense mutation (R210C) of the lysosomal/melanosomal ion channel TPC2 in a child with non-syndromic cutaneous albinism.

They used two complementary approaches to investigate the causal relationship. First, they studied the impact of R210C on the TPC2 current and its activation by phosphoinositides. Patch-clamp recording in an ectopic membrane (TPC2 sorting mutant) and in the endogenous organelle (enlarged lysosomes) showed that R210C increases the affinity of TPC2 for PI3,5P2 by almost two orders of magnitude, resulting in its constitutive activation. Second, they generated a TPC2 knock-in mouse line with the equivalent mutation, which showed hypopigmentation, decreased melanosome density in the retina, and decreased lysosomal pH in cultured MEFs.

The study is carefully designed and executed. It provides compelling evidence for the causative, gain-of-function role of TPC2 R210C in albinism, and supports a mechanism where the R210C mutation, in the phosphoinositide binding pocket of TPC2, induces its constitutive activation by basal levels of PI3,5P2 (and possibly other phosphoinositides), resulting in lysosome, and presumably melanosome, overacidification and impaired melanin synthesis, in agreement with the known role of melanosomal pH in other genetic forms of albinism.

Minor points:

1) Fig 1G and p. 6: the authors mention that the R210 side chain interacts with a fatty acid carbonyl of PI3,5P2 in the closed, but not the open, cryo-EM structure of TPC2. However, it is unclear how this interaction could account for the increased affinity for PI3,5P2, and the decreased head group selectivity, of the R210C mutant. This aspect should be discussed.

2) p. 13, line 275: The citation for the A1N compound should be ref. 18, not 24.

Reviewer #2:

Remarks to the Author:

The manuscript is based on the identification of a de novo TPCN2 variant from a young patient with fair skin and hair. The variant encodes a TPC2 ion channel with a point mutation, R210C, found in an intracellular loop along with other positive charges, three of which (K203, 204, and 207) have been involved in binding PI(3,5)P2. To test whether this mutation is the cause of the light skin/hair phenotype, the authors introduce the same mutation in the mouse homologue of TPCN2 and obtain a similar phenotype.

The rest of the experiments focus on identifying the mechanism by which the TPC2 R210C mutation causes the observed phenotypes in humans and mice. The authors use patch clamp of heterologous cells expressing a TPC2 mutant that localizes at the plasma membrane or endolysosomal TPC2 or mouse embryonic fibroblasts (MEF) from mutant or wild type mice. The data are suggestive of the conclusions of the manuscript, but fail to show it directly.

Overall this is an interesting manuscript describing a new TPC2 mutation affecting skin and hair pigmentation, but the proposed mechanism responsible for the phenotype is not fully supported by the data.

Major points:

1. In Fig. 2 the basal current for both WT and R210C TPC2 is negligible. However, the PI(3,5)P2 induced whole cell current is significantly smaller in HEK cells expressing the mutant TPC2, presumably due to lower expression of the mutant channel. However, a similar result could be due to smaller amplitude single channel TPC2 R210C currents or to the cells being smaller. Thus, the current

density should be plotted for all the experiments in which the channel is expressed in heterologous cells. It is also critical to show single channel recordings in order to argue that the regulation of the mutant channels is different.

2. The experiments in Figs 2 and 3 conclude that the TPC2 R210C mutant is constitutively active and thus a GOF mutation. But there is no direct evidence that this is the case. The concentration of PIP(3,5)P2 in endolysosomes is not known, so the right shifted dose response in Fig. 3C might not mean that the mutant TPC2 is active.

3. The conclusion that the mutant TPC2 R210C spontaneously localizes to the plasma membrane is based on one set of images (Fig. S2), in which the TPC2 localization is not quantified (and the resolution might be too low for proper quantification) and cells with vastly different sizes are shown in A and in B (if the calibration bar values in the legend are correct).

4. The baseline currents in Fig. 4 do not correspond to the currents shown in Fig. 2. There is no proof for the assumption that TPC2 R210C is constitutively active at the plasma membrane due to the presence of PI(3,5)P2, which is thought to be exclusively present in intracellular organelles.

5. Figure 5 shows a lysosomal TPC2 R210C current that is different than the plasma membrane current. Without single channel recordings it is difficult to conclude what causes the differences between the recordings shown in Fig 2 and Fig. 5.

6. The change in lysosomal calcium release (Fig. 7) is not convincing.

7. The observed phenotype consists of reduced melanin content in hair and skin. Because melanin is produced and stored in melanosomes, lysosomal related organelles, the authors conclude that the inferred hyperacidification of lysosomes will also happen in melanosomes and cause reduced activity of the enzyme that generates melanin. While this is a possibility, there is no data in the manuscript to show it.

Reviewer #1 (Remarks to the Author):

In this study, Wang et al report on a de novo missense mutation (R210C) of the lysosomal/melanosomal ion channel TPC2 in a child with non-syndromic cutaneous albinism. They used two complementary approaches to investigate the causal relationship. First, they studied the impact of R210C on the TPC2 current and its activation by phosphoinositides. Patch-clamp recording in an ectopic membrane (TPC2 sorting mutant) and in the endogenous organelle (enlarged lysosomes) showed that R210C increases the affinity of TPC2 for PI(3,5)P₂ by almost two orders of magnitude, resulting in its constitutive activation. Second, they generated a TPC2 knock-in mouse line with the equivalent mutation, which showed hypopigmentation, decreased melanosome density in the retina, and decreased lysosomal pH in cultured MEFs.

The study is carefully designed and executed. It provides compelling evidence for the causative, gain-of-function role of TPC2 R210C in albinism, and supports a mechanism where the R210C mutation, in the phosphoinositide binding pocket of TPC2, induces its constitutive activation by basal levels of PI(3,5)P₂ (and possibly other phosphoinositides), resulting in lysosome, and presumably melanosome, overacidification and impaired melanin synthesis, in agreement with the known role of melanosomal pH in other genetic forms of albinism.

We are very grateful for the reviewer's positive evaluation of the work's quality and significance. We are grateful to the family of the young girl who carries the TPC2^{R210C} mutation for their cooperation, which made it possible for the finding of this important new mechanism of TPC2-PI(3,5)P₂ regulation. Another key point of the current study is that the knock-in mice recapitulated the major clinical phenotypes of the patient, allowing us to investigate the potential cellular and organellar dysfunction likely underlying the mechanism of how the TPC2 gain-of-function mutant affects melanin production in melanocytes. There are certainly many interesting questions to address in future studies as we continue to work on TPC2.

Minor points:

1) Fig 1G and p. 6: the authors mention that the R210 side chain interacts with a fatty acid carbonyl of PI(3,5)P₂ in the closed, but not the open, cryo-EM structure of TPC2. However, it is unclear how this interaction could account for the increased affinity for PI(3,5)P₂, and the decreased head group selectivity, of the R210C mutant. This aspect should be discussed.

We added Supplementary Fig. 1b-g to show the likely structural changes in the TPC2^{R210C} mutation. The related description is provided in Results and Discussion sections of the revised manuscript.

2) p. 13, line 275: The citation for the A1N compound should be ref. 18, not 24.

This has been corrected in the revised manuscript.

Reviewer #2 (Remarks to the Author):

The manuscript is based on the identification of a de novo TPCN2 variant from a young patient with fair skin and hair. The variant encodes a TPC2 ion channel with a point mutation, R210C, found in an intracellular loop along with other positive charges, three of which (K203, 204, and 207) have been involved in binding PI(3,5)P2. To test whether this mutation is the cause of the light skin/hair phenotype, the authors introduce the same mutation in the mouse homologue of TPCN2 and obtain a similar phenotype.

The rest of the experiments focus on identifying the mechanism by which the TPC2 R210C mutation causes the observed phenotypes in humans and mice. The authors use patch clamp of heterologous cells expressing a TPC2 mutant that localizes at the plasma membrane or endolysosomal TPC2 or mouse embryonic fibroblasts (MEF) from mutant or wild type mice. The data are suggestive of the conclusions of the manuscript, but fail to show it directly.

Overall this is an interesting manuscript describing a new TPC2 mutation affecting skin and hair pigmentation, but the proposed mechanism responsible for the phenotype is not fully supported by the data.

We thank the reviewer for the careful assessment of our work and for finding the story interesting. We have performed additional experiments to address the reviewer's concerns, and the new results support our general conclusions.

Major points:

1. In Fig. 2 the basal current for both WT and R210C TPC2 is negligible. However, the PI(3,5)P2 induced whole cell current is significantly smaller in HEK cells expressing the mutant TPC2, presumably due to lower expression of the mutant channel. However, a similar result could be due to smaller amplitude single channel TPC2 R210C currents or to the cells being smaller. Thus, the current density should be plotted for all the experiments in which the channel is expressed in heterologous cells. It is also critical to show single channel recordings in order to argue that the regulation of the mutant channels is different.

We are very grateful to the reviewer's constructive and insightful comments. In the original Fig. 2a, we did not clearly point out that we performed inside-out recording of excised patches from cells that expressed a plasma membrane-localized TPC2^{L11A/L12A} mutant in the scheme. In the revised Fig. 2a, we have redrawn the scheme to emphasize this point. The currents shown in Figs. 2-4 are macroscopic currents from inside-out patches using pipettes of 6-8 MΩ resistance. For these recordings we do not trust

capacitance measurement for two reasons. The first and also the common reason for all inside-out patches is the poor correlation between capacitance measurement and the true membrane area included in the patch. Due to the negative pressure in the pipette, the membrane areas are distorted from the original membrane in size and shape. The second one is the low resistance of the membrane patch due to the constitutive activity of TPC2^{R210C}. Capacitance measurement is accurate only for high resistance membranes. Although we could estimate membrane areas based on the diameters of individual pipettes, these estimates have a high error rate (50% and above). Furthermore, since channel distributions on the plasma membrane are typically uneven, current density is really not more appropriate than current amplitude to present results from randomly selected small areas as for inside-out recordings, unlike whole-cell studies.

As for the basal current of TPC2-R210C^{L11A/L12A}, we always waited for a few minutes after excising the patch in order for the basal current to stabilize before applying PI(3,5)P₂. This is why the current immediately before PI(3,5)P₂ application was not so much larger in TPC2^{R210C} in Fig. 2e than the WT channel in Fig. 2c. Our goal here is to present the differences (kinetics and amplitude) in the PI(3,5)P₂-evoked activation and wash off between WT and TPC2^{R210C}.

The reviewer is absolutely right in that the smaller macroscopic current of the TPC2^{R210C} mutant could also result from a decrease in unitary current amplitude. We now include single channel data (new Fig. 5g-i) showing that TPC2^{R210C} increased open probability under unstimulated conditions without changing the single channel current amplitude. This should rule out the above possibility.

2. The experiments in Figs 2 and 3 conclude that the TPC2 R210C mutant is constitutively active and thus a GOF mutation. But there is no direct evidence that this is the case. The concentration of PIP(3,5)P₂ in endolysosomes is not known, so the right shifted dose response in Fig. 3C might not mean that the mutant TPC2 is active.

We appreciate the reviewer's comments and apologize for the confusion. Although constitutive activity represents one form of GOF, it is not the only one. The GOF of TPC2^{R210C} has at least three components: *a*) an increase affinity to PI(3,5)P₂, *b*) a gain of sensitivity to additional phosphoinositides, and *c*) constitutive channel activation. These may or may not be related. From Figs. 2 and 3, we concluded that TPC2^{R210C} is a GOF based on its faster response to PI(3,5)P₂ and slower wash-out (Fig. 2), and the large (92 fold decrease in EC₅₀) left shift of the concentration response curve (Fig. 3). All these point to an increased sensitivity and affinity to PI(3,5)P₂, which should already qualify the definition of GOF. We have reorganized the text in Results section to avoid directly linking GOF to constitutive activity.

The constitutive activation of TPC2^{R210C} is shown in Figs. 4 and 5. As described in the text for Fig. 4a, we found that unlike the TPC2^{L11A/L12A}-expressing cells, the majority (> 70%) of the TPC2^{L11A/L12A-R210C}-expressing cells failed to form a tight seal (>

GΩ) in cell-attached on-cell mode. This could indicate either non-specific leak or constitutive activity. The fact that after patch excision the basal current gradually decreased suggests that the apparent “poor seal” was due to constitutive activity supported by a labile agent(s). In the revised Fig. 5c-f, we further show that TPC2^{R210C} is constitutively active in endolysosomes with the constitutive currents blocked by NMDG⁺ (a positively charged large organic cation impermeable to TPCs) and three TPC2 antagonists, YM201636, tetrandrine and Ned19. Moreover, we detected single channel activity in inside-out patches excised from cells that expressed TPC2^{R210C}, which shows the same unitary amplitude as TPC2^{WT} (Fig. 5g-i). However, TPC2^{WT} did not show activity unless being delivered to the plasma membrane by the pretreatment of 30 μM A1N and stimulated by 100 nM PI(3,5)P₂ (see Materials and Methods section for details). Together, these data provide direct evidence that TPC2^{R210C} is constitutively active on both the PM and the endolysosomal vacuoles.

It is true that the concentrations of PI(3,5)P₂, as well as that of PI3P, PI5P, and PI(3,4,5)P₃, in endolysosomes and plasma membrane are not known, and probably vary from time to time and area to area. Therefore, we cannot conclude that the constitutive activity of the TPC2^{R210C} mutant results exclusively from the increased affinity to phosphoinositides. However, given the involvement of TPC2^{R210C} in PI(3,5)P₂ binding, the active turnover of PI(3,5)P₂, PI3P, PI5P, and PI(3,4,5)P₃ in the cell, and the easy washout of phosphoinositides (best known for PI(4,5)P₂) from inside-out patches by perfusion, the likelihood that one or several of the above phosphoinositides are responsible for the constitutive R210C activation is quite high.

3. The conclusion that the mutant TPC2 R210C spontaneously localizes to the plasma membrane is based on one set of images (Fig. S2), in which the TPC2 localization is not quantified (and the resolution might be too low for proper quantification) and cells with vastly different sizes are shown in A and in B (if the calibration bar values in the legend are correct).

We have added the quantification for the plasma membrane location in the revised Supplementary Fig. 2b. The original images were captured on a spinning disc microscope, so the resolution was high enough to quantify the fluorescent signal on the plasma membrane. The scale bars are provided in the revised Supplementary Fig. 2a and 2c. The single channel data for unstimulated TPC2^{R210C} in inside-out plasma membrane patches shown in revised Fig. 5g-i provide another independent support that TPC2^{R210C} spontaneously localizes to the plasma membrane.

4. The baseline currents in Fig. 4 do not correspond to the currents shown in Fig. 2. There is no proof for the assumption that TPC2 R210C is constitutively active at the plasma membrane due to the presence of PI(3,5)P₂, which is thought to be exclusively present in intracellular organelles.

As we have explained in point 1, unlike that in Fig. 4a, the currents shown in Fig. 2 did not start from the moment when the patch was excised. They started from the time when the current stabilized. In point 2, we explained our evidence on constitutive TPC2^{L11A/L12A-R210C} activity on the plasma membrane, which will not be repeated here. Although PI(3,5)P₂ is not known to be present on the plasma membrane, PI5P and PI(3,4,5)P₃ are abundantly found there. They could activate TPC2^{L11A/L12A-R210C} as well. Finally, the endogenous phospholipids have longer tails (C18 to C22) than the diC8 phosphoinositides used in our experiments, which may activate TPC2^{L11A/L12A-R210C} at even lower concentrations than what we report here.

5. Figure 5 shows a lysosomal TPC2 R210C current that is different than the plasma membrane current. Without single channel recordings it is difficult to conclude what causes the differences between the recordings shown in Fig 2 and Fig. 5.

The currents shown in Fig. 2 and Fig. 5 look different because in Fig. 2 we performed inside-out recordings using Na⁺-based solutions on both sides (see revised Fig. 2a), but in Fig. 5 the pipette contained mainly Na⁺ whereas the bath had mostly K⁺ (see Fig. 5b). The difference in solutions was emphasized in the revised manuscript.

6. The change in lysosomal calcium release (Fig. 7) is not convincing.

To address the reviewer's concern, we repeated this experiment for 3 more times by a different experimenter and obtained reproducible results. In addition, we used two different concentrations of A1N, 10 μM and 30 μM. We found that while 10 μM A1N caused Ca²⁺ release from lysosomes without coupling to endoplasmic reticulum (ER), 30 μM A1N induced Ca²⁺ release from both lysosomes and ER. This is not surprising as it is known that TPC2-mediated lysosomal Ca²⁺ release can be coupled to ER Ca²⁺ mobilization through the Ca²⁺-induced Ca²⁺ release mechanism¹. Therefore, we used 10 μM A1N to assess Ca²⁺ release from lysosomes in MEF cells, showing that the mTPC2-R194C knock-in cells produced the fastest and most robust cytosolic Ca²⁺ increase among the three genotypes (Fig. 7a-e). With 30 μM A1N, although the total cytosolic Ca²⁺ increase was comparable among the three genotypes because of the pronounced ER contribution, the Ca²⁺ rising phase started earlier and increased faster in mTPC2-R194C knock-in cells than in WT MEFs (Fig. 7f, g). These are all consistent with R194C being a GOF TPC2 mutant with the increased capability to release Ca²⁺ from lysosomes.

To investigate whether human TPC2^{R210C} has a similar effect on lysosomal Ca²⁺ release as mTPC2^{R194C}, we expressed TPC2^{R210C} with a GCaMP6 fused to its N-terminus (GCaMP6-TPC2^{R210C}) in HeLa cells. GCaMP6-TPC2^{WT} was used as a control. The attachment of the genetically encoded Ca²⁺ indicator to TPC2 N-terminus allows detection of [Ca²⁺] changes near the activated TPC2 at the cytoplasmic side. Similar to the findings in mTPC2^{R194C} knock-in

MEFs, A1N (30 μ M) caused a faster increase in GCaMP6-TPC2^{R210C} than GCaMP6-TPC2^{WT} fluorescence in the nominally Ca²⁺-free solution (Rebuttal Fig. 1a, b). That A1N is more effective at releasing Ca²⁺ from TPC2^{R210C}-expressing endolysosomes than TPC2^{WT}-expressing ones was also evident from the greater overall fluorescence increase in GCaMP6-TPC2^{R210C} than in GCaMP6-TPC2^{WT} based on the area under the curve (Rebuttal Fig. 1a, c). Thus, we here show that R210C is a GOF mutant which likely increases Ca²⁺ release from lysosomes. This data could be included in our future publication.

Rebuttal Figure 1. Enhanced TPC2^{R210C} Initiated Ca²⁺ Release Monitored by a TPC2 Fused Genetically Encoded Ca²⁺ Indicator GCaMP6. HeLa cells were transfected with GCaMP6-TPC2^{WT} or CaMP6-TPC2^{R210C} for 6 hours and additional 12 hours were allowed for protein expression. GCaMP6 imaging was performed on a Nikon inverted microscope (Eclipse Ti). The same solutions were used as the Fura2 assay. Images were captured through a GFP filter cube with central excitation wavelength at 480 nm and the emission one at 510 nm. Time interval was set at 1 s and data analysis was conducted in Excel (Microsoft 365) and GraphPad Prism 7.0.

(a) Time courses of GCaMP6 fluorescence changes ($\Delta F/F_{min}$) with time points of additions of 30 μ M A1N and 1 μ M Ionomycin (IM) indicated by grey and green arrows, respectively. Note that in the Ca²⁺-free ECS, GcaMP6-TPC2^{R210C} exhibited a quicker and more robust

response to A1N than GCaMP6-TPC2^{WT}.

(b) Statistics of half-time to peak in response to A1N. GCaMP6-TPC2^{R210C} released calcium faster than GCaMP6-TPC2^{WT}. GCaMP6-TPC2^{WT}: 60.27 ± 0.64 s, n = 77; GCaMP6-TPC2^{R210C}: 42.84 ± 0.45 s, n = 83.

(c) Statistics of area under the trace between 120 s to 380 s, reflecting the overall lysosomal Ca²⁺ release caused by TPC2 activation via A1N. GCaMP6-TPC2^{WT}: 126.8 ± 4.6 , n = 77; GCaMP6-TPC2^{R210C}: 159.1 ± 3.7 , n = 83. RFU, random fluorescence unit.

*** $p < 0.001$ determined by unpaired t -test.

7. The observed phenotype consists of reduced melanin content in hair and skin. Because melanin is produced and stored in melanosomes, lysosomal related organelles, the authors conclude that the inferred hyperacidification of lysosomes will also happen in melanosomes and cause reduced activity of the enzyme that generates melanin. While this is a possibility, there is no data in the manuscript to show it.

Melanosomal pH measurement and melanosomal patch clamp recording remain to be the major technical obstacles in this field. Chemical pH indicators, such as LysoSensor, are not suitable for melanosomal pH measurement because a) they do not specifically target to the melanosomes, and b) melanin quenches their fluorescence. A genetically encoded melanosome-localized pH sensor, MELOPS, was created and used for melanosomal pH measurement in MNT-1 melanocytic cells and primary human melanocytes before², however, in our hands, when transfected in MNT-1 or B16F10 cells, MELOPS failed to localize to melanosomes and caused severe cell death. In fact, all the transfected cells displayed a bright fluorescence aggregate and rounded up (data not shown). Therefore, we had to measure lysosomal pH to infer the pathogenesis of hypopigmentation. In previous publications, increases in TPC2 or ClC-7 function and a loss-of-function of OCA2 were referred to lower lysosomal or melanosomal pH, which were well accepted as the reason for hypopigmentation²⁻⁵.

References

1. Calcraft, P. J., *et al.* NAADP mobilizes calcium from acidic organelles through two-pore channels. *Nature* **459**, 596–600 (2009)
2. Ambrosio, A. L., Boyle, J. A., Aradi, A. E., Christian, K. A. & Di Pietro, S. M. TPC2 controls pigmentation by regulating melanosome pH and size. *Proceedings of the National Academy of Sciences* **113**, 5622 (2016).
3. Bellono, N. W., Escobar, I. E. & Oancea, E. A melanosomal two-pore sodium channel regulates pigmentation. *Sci Rep* **6**, 26570 (2016).
4. Nicoli, E.-R. *et al.* Lysosomal Storage and Albinism Due to Effects of a De Novo CLCN7 Variant on Lysosomal Acidification. *Am J Hum Genet* **104**, 1127–1138 (2019).
5. Bellono, N. W., Escobar, I. E., Lefkovith, A. J., Marks, M. S. & Oancea, E. An

intracellular anion channel critical for pigmentation. *Elife* **3**, e04543 (2014).

Reviewers' Comments:

Reviewer #1:

Remarks to the Author:

The authors satisfactorily addressed all concerns.

Reviewer #3:

Remarks to the Author:

The authors have responded to the previous reviewers' comments in detail and this reviewer cannot add to these comments.

The genetic analysis is convincing and the candidacy of TPC2 is supported by much prior work. The mutation described, R210C, has been previously seen, albeit extremely rarely, in large scale population sequencing projects and has the SNP id of rs540210379. The individual with the variant was from a South Asian population. Although there has been no reported association with a pigmentation phenotype this is not surprising in a large scale sequencing programme. Uniprot assigns this variant as "probably damaging".

The phenotype of the mouse model is surprisingly mild; there might be a stronger phenotype on an agouti background.

An additional, fairly minor point, is the analysis of melanin in the eye. Figure 6g is labelled "relative RPE melanin content". In fact the excised eye will contain not only the melanin but also the choroid, which is heavily pigmented. The difference between wild-type and mutant mouse eyes in melanin content is small, whilst the melanosome density difference in Figure 6h is rather larger. If they only measured melanosome density in the RPE then it may be (as a superficial look at the TEM sections suggests) there is a bigger effect in the RPE compared to the choroid. The authors should make clear what Figures 6g and 6h are measuring.

Reviewer #3 (Remarks to the Author):

The authors have responded to the previous reviewers' comments in detail and this reviewer cannot add to these comments.

Reply: We are very thankful for the reviewer's comments.

The genetic analysis is convincing and the candidacy of TPC2 is supported by much prior work. The mutation described, R210C, has been previously seen, albeit extremely rarely, in large scale population sequencing projects and has the SNP id of rs540210379. The individual with the variant was from a South Asian population. Although there has been no reported association with a pigmentation phenotype this is not surprising in a large scale sequencing programme. Uniprot assigns this variant as "probably damaging".

Reply: We appreciate the reviewer's comments. *TPCN2* (c.628C>T, p.Arg210Cys, dbSNP: rs540210379) was predicted as a "probably damaging" mutation by multiple predictors as listed below:

Polyphen// probably_damaging, SIFT// deleterious, Uniprot// probably_damaging.

This supports our findings that the mutation could be pathogenic as we mentioned in lines 101-102 in our original manuscript. However, this mutation, has not previously been linked to hypopigmentation, and has not been listed in the two common disease-causing mutation databases, HGMD (<https://www.hgmd.cf.ac.uk/docs/login.html>) and ClinVar (<https://www.ncbi.nlm.nih.gov/clinvar>). In addition, this variant was found in only one case in our local albinism database that contains the whole exome sequences of more than 2000 albinism patients. In this study, we used functional assays to verify the pathogenicity of *TPCN2*-R210C both *in vivo* and *in vitro*. This is the first time to link the genotype to hypopigmentation.

The phenotype of the mouse model is surprisingly mild; there might be a stronger phenotype on an agouti background.

Reply: Our knock-in mouse was generated in the non-agouti C57Bl/6J background which may make the color changes not so profoundly. We may consider to transfer this mutation in an agouti mouse background to see if the coat color change will be more prominent in future.

An additional, fairly minor point, is the analysis of melanin in the eye. Figure 6g is labelled "relative RPE melanin content". In fact the excised eye will contain not only the melanin but also the choroid, which is heavily pigmented. The difference between wild-type and mutant mouse eyes in melanin content is small, whilst the melanosome density difference in Figure 6h is rather larger. If they only measured melanosome density in the RPE then it may be (as a superficial look at the TEM sections suggests) there is a bigger effect in the RPE compared to the choroid. The authors should make clear what Figures 6g and 6h are measuring.

Reply: Thanks for raising this point. In the revised version, we change the Y label of Figure 6g to “Relative Melanin Content”. The corresponding figure legends have been revised as “Relative melanin contents in eye tissues were $82.3 \pm 2.0\%$ (Het, n = 5) and $83.8 \pm 3.8\%$ (Hom, n=6) of that of WT mice”.

We calculated the melanosome density in the choroid sections. We found there is no significant difference between these three genotypes (WT, Het, Hom. See the new Figure 6i included in the revised Figure 6). These results suggest that TPC2 may be involved in the biogenesis of melanosomes in the RPE rather than in the choroid as the melanocytes of these two regions originate differently during embryonic development and their melanosomal compositions are different [1]. Thus, the reduced number of RPE melanosomes may contribute to the reduction of melanin content (relatively mild), which is buffered by the normal melanin content from choroidal melanocytes as we measured the melanin content in the whole eyes. However, we did not see much difference in the colors of iris and retina from the patient, which was documented in our original manuscript.

References

1. Ward, W.C. & Simon, J.D. The differing embryonic origins of retinal and uveal (iris/ciliary body and choroid) melanosomes are mirrored by their phospholipid composition. *Pigment Cell Res* **20**, 61-69 (2007).